

# Insights into isotopic mismatch between soil water and *Salix matsudana* Koidz xylem water from root water isotope measurements

Ying Zhao[1,2], Li Wang[1,3]

[1]State Key Laboratory of Soil Erosion and Dryland Farming on the Loess Plateau, Institute of Soil and Water Conservation,
Chinese Academy of Sciences and the Ministry of Water Resources, Yangling 712100, China
[2]University of Chinese Academy of Sciences, Beijing 100049, China
[3]College of Natural Resources and Environment, Northwest A&F University, Yangling 712100, China

*Correspondence to*: Li Wang (wangli5208@nwsuaf.edu.cn)

**Abstract.** Increasing numbers of field studies have detected isotopic mismatches between plant xylem water and its potential sources. However, the cause of these isotopic offsets is not clear and it is uncertain whether they occur during root water uptake or during water transmission from root to xylem. Thus, we measured the specific isotopic composition ($\delta^2H$ and $\delta^{18}O$) of soil water, groundwater, xylem water, and root water of *Salix matsudana* Koidz trees at high temporal resolution to analyze isotopic dynamics in the soil-root-xylem continuum. We report three main findings. First, we detected clear separation between mobile water and bulk soil water isotopic signals, supporting the 'two water worlds' (TWW) hypothesis. Second, the isotopic composition of bulk soil water was closest to and overlapped with that of root water at 0-60 cm depths, but $\delta^2H$ and $\delta^{18}O$ values of root water at 80-160 cm depths deviated significantly from that of bulk soil water at the root-soil interface. This was likely due to separation of mobile and tightly bound soil water (as in the TWW hypothesis) and plant fractionation. The maximum differences in $\delta^2H$ and $\delta^{18}O$ between bulk soil water and root water were −8.6 and −1.8‰, respectively. Third, xylem water was only isotopically similar to root water at 100-160 cm depths and these root layers provided 74% of the xylem water. In conclusion, isotopic offset occurred at the interface between the soil and *S. matsudana* roots, and it can be attributed to a combination of plant fractionation and TWW-type separation of bound and mobile soil water. Our study contributes to the body of knowledge on isotopic dynamics in the soil-root-xylem continuum and provides potentially valuable insights regarding isotopic offsets between soil water and xylem water of *S. matsudana* tree and other species in similar conditions.

## 1 Introduction

Root water uptake (RWU) is the main mechanism through which plants obtain the water they require for photosynthesis, metabolism and maintenance (McCormack et al., 2015). RWU also controls partitioning of infiltrated soil water between groundwater recharge and local atmospheric return through evapotranspiration (Knighton et al., 2020a; Knighton et al., 2020b), and thus plays a key role in global hydrological cycles. In terrestrial ecosystem, plant transpiration accounts for more than 60% of total evapotranspiration and returns approximately 39% of incident precipitation to the atmosphere (Schlesinger and Jasechko, 2014; Good et al., 2015). However, although the pivotal role of RWU has been long recognized, there is limited





understanding and quantification of RWU because of the opaque nature of the soil and variability in time and space of the RWU process.

Analyses of isotopes in water ($\delta^{18}O$ and $\delta^2H$) have been extensively applied in attempts to determine the sources of water used by plants, providing useful insights into the RWU process (Rothfuss and Javaux, 2017; Penna et al., 2020). This application relies on the assumption that RWU is generally a non-fractionating process (Ehleringer and Dawson, 1992), so the isotopic composition of xylem water effectively reflects that of water sources. Thus, by comparing $\delta^{18}O$ and $\delta^2H$ values of plant xylem water to those of potential contributory water sources (e.g., water from different soil layers, groundwater and precipitation), the relative contributions of these water sources to RWU can be estimated (Rothfuss and Javaux, 2017; Wang et al., 2018; Wang et al., 2020). However, a growing body of evidence indicates that there is an isotopic offset between xylem water and potential plant water sources, that is, the isotopic composition of xylem water does not match any of the considered water sources in the dual-isotope space (Bowling et al., 2017; Vargas et al., 2017; Barbeta et al., 2019). This implies that water isotope composition changes in the movements from soil to root and then to xylem, which might not be solely due to isotopic fractionation (Poca et al., 2019), but also to ecohydrological separation (Brooks et al., 2010), water isotope heterogeneity (Barbeta et al., 2020). Observed offsets could also be at least partly due to the water extraction technology used (Orlowski et al., 2018). The contributions of these factors to the isotopic deviations are uncertain.

Analyses of isotopic signals ($\delta^2H$ and $\delta^{18}O$) within watersheds have suggested that groundwater is isolated from water sources used by plants, a phenomenon called ecohydrological separation or the 'two water worlds' (TWW) hypothesis (Brooks et al., 2010; McDonnell, 2014). This hypothesis is broadly supported on a global scale by enrichment of $\delta^2H$ and $\delta^{18}O$ values in soil water and xylem water, but not groundwater and streams (Evaristo et al., 2015). The isotopic offset between plant xylem water and groundwater has been attributed, at least in some areas, to two water pools in the soil matrix: a tightly bound water pool used by plants and a mobile water pool related to infiltration and groundwater recharge via preferential flow (Evaristo et al., 2016; Bowling et al., 2017). The TWW hypothesis relies on the mentioned assumption that no isotopic fractionation occurs during RWU, but some studies indicate that such fractionation probably does contribute to the isotopic offset (Vargas et al., 2017; Barbeta et al., 2019). For example, Lin and Sternberg (1993) and Ellsworth and Williams (2007) found evidence that hydrogen isotope fractionation occurs during RWU of halophytic and xerophytic plants. In addition, Poca et al. (2019) reported that arbuscular mycorrhizal fungi can enhance isotope fractionation during RWU, resulting in up to $-24.6‰$ and $-2.9‰$ differences in $\delta^2H$ and $\delta^{18}O$ values, respectively, between soil and plant xylem water. However, Barbeta et al. (2020) concluded that isotopic mismatches between soil water and xylem water are less likely to be caused by plants' fractionation than by water isotope heterogeneities in plant tissues and soil pores. In addition, effects of extraction technology must be considered. For example, incomplete extraction of water during cryogenic distillation could fractionate water isotopes (Gaj et al., 2017; Orlowski et al., 2018). Chen et al. (2020) found the common presence of significant isotopic deviations between stem water cryogenically extracted and source water in nine woody plant species and demonstrated that this offset stems from a cryogenic extraction-associated methodological artifact. Thus, the extracted water does not properly represent the water available to plants, and may contribute to apparent xylem-soil water isotopic offsets.





Explanation of the isotopic offset between soil and xylem water is essential, but identifying roles of specific process is generally hindered by the diversity of mechanisms that may be involved (Sprenger and Allen, 2020). Moreover, these mechanisms tend to have strongly interactive effects and may act on any compartment along soil-root-xylem continuum, leading to the variation in water isotopes. In addition, plants' roots transmit water from soil to xylem, and thus may play key roles in isotopic variation, e.g., root preferentially use tightly bound water according to the TWW hypothesis (Brooks et al.,

2010) and mycorrhizal fungi may contribute to fractionation (Poca et al., 2019). However, much more attention has been paid to the isotopic composition of plant xylem water and potential water sources (Chang et al., 2019; Kuhnhammer et al., 2020) than to isotopic signatures of root water due to the inaccessibility of roots (Zhao et al., 2016), leading to a lack of key information to explain observed mismatches.

Therefore, the aim of the study presented here was to analyze hydrogen and oxygen isotope composition dynamics in the

water transport process along the soil-root-xylem continuum. More specifically, we exploited the specific isotope fingerprints ($\delta^2$H and $\delta^{18}$O values) of mobile water, bulk soil water, groundwater and xylem water of *Salix matsudana* trees, and derived line-conditioned excess (lc-excess) values to assess the TWW hypothesis. We compared the isotopic composition of root and soil water at root-soil interface at 0-160 cm depths, as well as the isotopic composition of root and xylem water during water transport from root to xylem, to identify more specifically the sites and causes of the isotope deviation. Finally, we used the

SIAR model to calculate contributions of root water and soil water to xylem water. We hypothesize that there is an isotopic deviation between xylem water of *S. matsudana* trees and their potential water sources (the first hypothesis), and that this deviation might be due to a combination of multiple factors (the second hypothesis).

## 2 Materials and methods

### 2.1 Site description

The study was conducted in the Liudaogou catchment (38°47′-38°49′N, 110°21′-110°23′E) on Loess Plateau of China (Fig. 1). The area and altitude of the catchment are 6.89 km$^2$ and 1081-1274 m, respectively. The regional climate is classified as semi-arid with cool dry winters and most of precipitation occurs during the warm summer season. The mean annual precipitation and temperature in the catchment are 464 mm and 8.4 ℃, respectively. The study area received less than usual precipitation (426 mm) in the study year (2019). During this year, the seasonal distribution of precipitation was uneven, mostly

concentrated in July to September (70%), and the average daily temperature ranged from −13.48 ℃ in January to 26.20 ℃ in July. The Liudaogou catchment is in the 'water-wind erosion crisscross area' of the plateau. The soil erosion modulus for this area is reportedly 15,040 t km$^{-2}$ a$^{-1}$ (Gong et al., 2018). Severe soil erosion has caused strongly fragmented landforms, with gullies accounting for ca. 38% of the total area (Zhu and Shao, 2008). Both vegetation and engineering measures (check dams) are used here to mitigate soil erosion. Common species used in reforestation of the area include *Salix matsudana* Koidz, *S.*

*psammophila*, *Caragana korshinskii* and *Medicago sativa* L. Check dams are usually built in gullies and other channels in the area to trap runoff and sediments from steep slopes and improve agricultural yields.



**Fig. 1**

We selected three sampling sites in the check-dammed channel of the Liudaogou catchment, 50, 80 and 100 m upstream of the dam, designated sites 1, 2 and 3, respectively (Fig. 1). *Salix matsudana* Koidz, one of the main tree species in the check-
dammed catchment, was chose for sampling tree. The average age and height of the trees are about 30 years and 12 m, respectively. The soil at the site includes sandy loam and loam according to the USDA classification system (Table 1) and its bulk density ranges from 1.4 to 1.6 g cm$^{-3}$. Water retention curves of different soil layers in a sampling plot are shown in Fig. S1. Meteorological data on precipitation and air temperature (with 30-min resolution) were obtained from a weather station located about 500 m from the sampling plot. Precipitation was measured using TE525 rain gauges (Campbell Scientific Inc.),
which provide ± 1 percent accuracy at rates up to 25.4 mm hr$^{-1}$. Air temperature was measured using HMP45D probes, which have ± 0.2 °C accuracy at 20 °C (Vaisala Inc.).

**Table 1**

## 2.2 Measurements of roots and soil properties

We collected the root samples of *S. matsudana* tree at each sampling site, on August 18, 2019, to measure roots' isotopic
composition. We excavated a soil cuboid with 160 cm depth, 80 cm width (horizontal distance) and 160 cm length with the main root of the selected tree at the center (Fig. 1d). We then divided the cuboid into 64 sub-cuboids (length: 40 cm, width: 40 cm, height: 20 cm) (Fig. 1d). The coarse roots (> 2 mm in diameter) in each sub-cuboid was collected and measured its isotopic composition. We also collected disturbed soil samples at 0-160 cm depths using a soil auger to measure soil particle size, and undisturbed soil samples at 20, 30, 50, 100 and 150 cm depths using cutting rings (100 cm$^3$ in volume). We then took the
samples to the laboratory and determined their particle size and bulk density using a MS 2000 Laser Particle Size Analyzer (Malvern Instruments, Malvern, UK), and obtained water retention curves for them using a CR21G high-speed centrifuge (Hitachi, Japan).

## 2.3 Water sampling for stable isotope ($\delta^2$H and $\delta^{18}$O) analysis

Our previous results have shown that the isotopic composition of xylem water of *S. matsudana* tree did not match soil
water in the dual-isotope space from May to September 2018 (Fig. S2). Thus, to assess the TWW hypothesis that separation of the bound water used by plants and mobile water strongly contributes to isotopic deviation (Brooks et al., 2010), we collected mobile and bulk soil water in 2019. Due to effects of drought, mobile water samples could not be obtained continuously from May to July 2019 (Table S1). So, high frequency sampling (ca. 3-day temporal resolution) was applied to analyze the causes and locations of isotope deviation within the period (August 4 to September 15 2019) when mobile water was available. Soil
water from 0-160 cm depths (bulk soil water, N=247; mobile water, N=191), groundwater (N=22), plants' xylem water (N=61) and root water (N=156) were collected for the hydrogen and oxygen isotopic analyses. For these analyses, precipitation samples were collected as soon as a rain event ended from a polyethylene funnel and bottle, with a plastic ball placed in the funnel to reduce evaporation. Groundwater samples were collected at a water well located about 300 m from the soil and root



sampling plot. Soil samples were collected at 10 cm intervals from 0-100 cm depths and 20 cm intervals from 100-160 cm

depths. The soil samples of each layer were divided into two groups: one for isotopic analysis and the other for determination of gravimetric soil water content (GWC, %) by the drying method (105℃ for 12 h). In parallel, mobile water was sampled at 20, 30, 50, 100 and 150 cm depths using suction lysimeters when water was present. Each lysimeter consisted of a porous cup with two inserted tubes that allowed creation of the vacuum in the lysimeter and sampling of soil water by injecting air into the lysimeter (Fig. 1b). A tension of 60 kPa was applied to each suction lysimeter.

Tree samples were collected simultaneously with the soil sampling campaign. These consisted of twigs collected from the south-facing side of three *S. matsudana* trees at 250 cm height on each sampling occasion. Bark was peeled from the twigs and all leaves were removed to avoid perturbance of xylem water isotopic signatures by fractionation. Pieces of the de-barked and de-leaved twigs, 30 mm long, were then immediately placed in 10 mL vials and wrapped in parafilm. In addition, samples of xylem at selected tree heights (150, 250, 350, 450 cm) and root samples at selected soil depths (0-160 cm with 20 cm

intervals) and horizontal distances (0-80 cm with 40 cm intervals, excavated as described in Section 2.2) were collected on August 18, 2019. Similarly, of 30 mm long pieces of the de-barked twigs were immediately placed in 10 mL vials and wrapped in parafilm.

## 2.4 Stable isotope analysis

A LI-2100 automated vacuum distillation system (LICA Inc., Beijing, China) was used to extract water from the soil,

xylem and root samples. This system is similar to cryogenic vacuum distillation systems that are widely used elsewhere (Gaj et al., 2017), except that it uses a compressor refrigeration unit and not liquid nitrogen. Samples were subjected to the maximum allowed vacuum pressure of 1500 Pa and temperature differential of 225 ℃ (heating temperature, 130 ℃; cooling trap, −95 ℃) for 180 min during extraction, in efforts to ensure that more than 99% of the water was collected from them. The $\delta^2H$ and $\delta^{18}O$ values for all samples were determined using an Isoprime 100 Stable Isotope Ratio Mass Spectrometer (Isoprime Ltd Inc.,

Cheadle, UK) at the Institute of Water-saving Agriculture in Arid Areas of China, Northwest A&F University. The precision of the analyses of H and O isotopes was 0.5 and 0.1‰, respectively. The isotopic composition ($^2H$ to $^1H$ and $^{18}O$ to $^{16}O$ ratios) of the samples was normalized relative to the V-SMOW (Vienna Standard Mean Ocean Water) standard set by the International Atomic Energy Agency. The resulting ratios were then expressed in delta notation ($\delta^2H$ and $\delta^{18}O$ values), calculated as follows:

$$\delta^2H(‰) = (\frac{R_{sample}}{R_{standard}}) - 1 \tag{1}$$

$$\delta^{18}O(‰) = (\frac{R_{sample}}{R_{standard}}) - 1 \tag{2}$$

## 2.5 Methods for assessing ecohydrological separation and determining plant water sources

To test the TWW hypothesis, we calculated the line-conditioned excess (lc-excess) values (Landwehr and Coplen, 2006) of bulk soil water, mobile water, groundwater and xylem water. The lc-excess values were used to identify the degree of 'offset' of environmental waters from precipitation. A negative lc-excess that exceeds the standard deviation of the local meteoric





water line (LMWL) indicates that water has undergone evaporative isotopic enrichment (Evaristo et al., 2016). The lc-excess values of samples were calculated as follows:

$$lc - excess = \delta^2 H_s - a\delta^{18}O_s \tag{3}$$

where the subscript 's' represents the sample and *a* and *b* are the slope and intercept of the LMWL, respectively. The LMWL shows the relationship between $\delta^2 H$ and $\delta^{18}O$ in precipitation, and according to analysis of the precipitation (N=89)

from 2016 to 2019 at our study site, this was $\delta^2 H = 7.67 \, \delta^{18}O + 5.91$.

In addition, following Sprenger et al. (2019), we estimated $\delta^2 H$ and $\delta^{18}O$ values of tightly bound water to test the TWW hypothesis. We determined the maximum value of tightly bound water at selected depths (20, 30, 50, 100 and 150 cm), that is, the GWC determined by application of 60 kPa suction (the tension applied to obtain mobile water). The mobile fraction of soil water was calculated from the difference between the measured bulk soil water and tightly bound water. Based on an isotope

mass balance approach, the isotopic composition of tightly bound water was calculated as follows:

$$\delta_{\text{TW}} = \frac{\delta_{\text{BW}} \cdot \theta_{\text{BW}} - \delta_{\text{MW}} \cdot \theta_{\text{MW}}}{\theta_{\text{TW}}} \tag{4}$$

Here, $\delta$ and $\theta$ represent the isotopic composition and GWC of samples, respectively, while, the subscripts 'TW', 'BW' and 'MW' represent tightly bound water, bulk soil water and mobile water, respectively.

To compare the isotopic composition of root and soil water at the same depth, we calculated contributions of root water

and soil water to xylem water of *S. matsudana* tree. For this, we employed Stable Isotope Analysis in R (SIAR, version 4.2) to quantify the sources of water taken up by the trees. This is a package designed to solve mixing models for isotopic data in a Bayesian framework (Parnell et al., 2010). Based on results of the soil water and root water isotopes analyses ($\delta^2 H$ and $\delta^{18}O$ values) (Fig. 5), soil water sources available for root uptake were divided into four categories when running the SIAR model: groundwater and soil water at 0-60, 60-100, 100-160 cm depths. In parallel, root water sources available for root uptake were

divided into the same four categories when running this model. In the SIAR model, the trophic enrichment factor was set to 0 for both $\delta^2 H$ and $\delta^{18}O$ because plant water use does not generally cause fractionation of hydrogen and oxygen isotopes (Ehleringer and Dawson, 1992). This model was run with 500,000 iterations (discarding the first 50,000 iterations) and the most likely contribution (mean of the posterior distribution) of a water source (root water or soil water) to xylem water of *S. matsudana* trees on August 18, 2019 was obtained.

**2.6 Statistical analysis**

All statistical analyses were performed using SPSS 20.0 (SPSS Inc., Chicago, USA). Shapiro-Wilk and Levene's tests were respectively used to check that the data met normality of distribution and homogeneity of variance requirements for planned analyses. One-way ANOVA followed by Tukey's test was used to detect significant differences in the variation in depth of soil water, root water, mobile water and bulk soil water isotopic composition. Presented diagrams were generated

using SigmaPlot 12.5.





## 3 Results

### 3.1 Dual-isotope plots

Plots of the stable isotopic composition ($\delta^2$H and $\delta^{18}$O) of all water samples in dual isotope space are shown in Fig. 2a. The slope and intercept of the local meteoric water line (LMWL, $\delta^2$H = 7.67 $\delta^{18}$O + 5.91, $R^2$ = 0.96) were lower than those of the global meteoric water line (GMWL, $\delta^2$H = 8 $\delta^{18}$O + 10) (Craig, 1961). Mobile water at 20, 30 and 50 cm depths ($\delta^2$H = 6.70 $\delta^{18}$O – 2.10, $R^2$ = 0.998, N = 113, $p < 0.001$) typically plotted within the 95% confidence interval of precipitation (Fig. 2b), while mobile water at 100 and 150 cm depths ($\delta^2$H = 5.91 $\delta^{18}$O – 13.05, $R^2$ = 0.94, N = 73, $p < 0.001$) slightly deviated from the LMWL. Groundwater was isotopically similar to mobile water at 150 cm depth. Bulk soil water ($\delta^2$H = 7.25 $\delta^{18}$O – 4.35, $R^2$ = 0.95, N = 199, $p < 0.001$) partly overlapped isotopically with mobile water but it generally plotted below mobile water (Fig. 2a and c). Root water ($\delta^2$H = 5.47 $\delta^{18}$O – 21.46, $R^2$ = 0.33, N = 156, $p < 0.001$) strongly deviated from the LMWL and was generally isotopically enriched compared to mobile water and bulk soil water except that root water at 0-60 cm depths overlapped with bulk soil water (Fig. 2d). Neither mobile water nor bulk soil water matched the isotope composition of xylem water. Xylem water ($\delta^2$H = 3.69 $\delta^{18}$O – 38.20, $R^2$ = 0.31, N = 61, $p < 0.001$) was only isotopically similar to root water at 100-160 cm depths (Fig. 2e).

**Fig. 2**

### 3.2 The lc-excess of mobile water and bulk soil water

**Fig. 3**

As shown in Fig. 3, the mean lc-excess values were −1.27 ± 2.10, −2.35 ± 0.62, −6.72 ± 1.24 and −13.82 ± 1.01 for mobile water, groundwater, bulk soil water and xylem water, respectively. The lc-excess values of groundwater and mobile water did not significantly differ ($p > 0.05$), and they were significantly higher than those of bulk soil water and xylem water ($p < 0.05$), indicating the existence of isotopic separation and supporting the TWW hypothesis. The lc-excess value of bulk soil water was generally higher than that of xylem water ($p < 0.05$), suggesting that xylem water was isolated from all potential water sources. This hypothesis is corroborated by tightly bound water data (Fig. S3), as lc-excess of xylem water is generally within the range of lc-excess of tightly bound water, indicating that plants might preferentially use tightly bound water.

During the sampling period (August 4 to September 15, 2019), GWC at 20 cm depth was strongly affected by precipitation and evaporation, ranging from 7.4 to 20.8% (mean: 11.9%). The lc-excess values of bulk soil water and mobile water were always significantly different under all soil moisture conditions ($p < 0.05$). GWC at 150 cm depth remained relatively stable, ranging from 9.3 to 15.3% (mean: 11.7%). There was a significant isotopic difference between mobile and bulk soil water in this layer ($p < 0.05$), but it was smaller than the corresponding difference in other soil layers. No correlation between Δlc-excess (lc-excess difference between mobile water and bulk soil water) and GWC was detected at 20-150 cm depths. At every sampling depth, the lc-excess of mobile water was always higher than that of bulk soil water ($p < 0.05$), but the Δlc-excess was most pronounced in the 20-50 cm depths. A strong positive correlation between lc-excess value and GWC was observed



at 20, 30 and 50 cm depth for mobile water (20 cm, $y = 0.19x - 1.27$, $R^2 = 0.27$, N = 40, p = 0.001; 30 cm, $y = 0.17x - 1.61$, $R^2$ = 0.22, N = 40, p = 0.002; 50 cm, $y = 0.20x - 3.59$, $R^2 = 0.16$, N = 38, p = 0.013) and 20-30 cm depths for bulk soil water (20

cm, $y = 0.34x - 11.39$, $R^2 = 0.30$, N =42, p < 0.001; 30 cm, $y = 0.23x - 9.21$, $R^2 = 0.20$, N =40, p = 0.003) (Fig. 4a-c). While, no correlation between these variables was detected (Fig. 4d and e) at 100 and 150 cm depths for mobile water and for bulk soil water.

**Fig. 4**

### 3.3 Comparison between root water and bulk soil water isotopes at different depths

**Fig. 5**

As shown in Fig. 5b and d, there were no significant differences (p > 0.05) in isotopic composition ($\delta^2$H and $\delta^{18}$O) of either root water or bulk soil water between 40 cm and 80 cm distances, suggesting that isotopic composition of the soil was horizontally homogenous > 80 cm from tap roots. However, isotopic variations with depth were detected in both root water and bulk soil water. Generally higher $\delta^2$H and $\delta^{18}$O values in root water (mean values and standard deviations for three soil

profiles: $-65.90 \pm 2.92$ and $-7.66 \pm 0.40$‰, respectively) than in bulk soil water (mean values and standard deviations for three soil profiles: $-69.09 \pm 2.50$ and $-8.89 \pm 0.38$‰, respectively) were observed at 80-160 cm depths. Root water and bulk soil water significantly differed in $\delta^2$H at 80-140 cm depths (p < 0.05) and $\delta^{18}$O at 60-160 cm depths (p < 0.01) (Fig. 5a, c). Although $\delta^2$H and $\delta^{18}$O values of root water and bulk soil water behaved differently, a strong correlation was observed between $\Delta^{18}$O ($\Delta^{18}$O = $\delta^{18}$O$_{soil}$ − $\delta^2$O$_{root}$) and $\Delta^2$H ($\Delta^2$H = $\delta^2$H$_{soil}$ − $\delta^2$H$_{root}$) for soil-root offset (Fig. 6a) at 0-160 cm depths (bulk soil

water-root water: $y = 3.83x + 0.99$, $R^2 = 0.69$, N = 24, p < 0.001). Similarly, a strong correlation was observed between $\Delta^{18}$O ($\Delta^{18}$O = $\delta^{18}$O$_{soil}$ − $\delta^2$O$_{xylem}$) and $\Delta^2$H ($\Delta^2$H = $\delta^2$H$_{soil}$ − $\delta^2$H$_{xylem}$) soil-xylem offsets during August 4 to September 15 (bulk soil water-xylem water: $y = 6.80x + 6.52$, $R^2 = 0.83$, N = 42, p < 0.001; mobile water-xylem water: $y = 5.93x + 10.87$, $R^2 = 0.81$, N = 42, p < 0.001) (Fig. 6b). These results show that water isotopes, especially hydrogen isotopes, changed between root water and soil water, and between soil water and xylem water, supporting our first hypothesis.

**Fig. 6**

### 3.4 Contributions

**Fig. 7**

Potential sources of plant xylem water were determined using a Bayesian mixing model approach. Fig. 7 shows contributions of potential water sources calculated from root water and bulk soil water isotopic signatures. According to root

water isotopic data, root water at 0-60, 60-100, and 100-160 cm depths and groundwater accounted for $3 \pm 3\%$ (mean ± 1SD), $17 \pm 9\%$, $74 \pm 10\%$ and $6 \pm 4\%$ of xylem water, respectively. According to bulk soil water isotopic data, bulk soil water at 0-60, 60-100, 100-160 cm depths and groundwater accounted for $26 \pm 6\%$, $32 \pm 10\%$, $33 \pm 12\%$ and $9 \pm 8\%$ of xylem water, respectively. Such large differences in contributions also suggest that an isotopic offset occurs at the root-soil interface.



## 4 Discussion

### 4.1 Isotopic dynamics at the root-soil interface

#### 4.1.1 Separation of mobile water and bulk soil water in the soil matrix

At our study site during the covered period (August 4 to September 15, 2019), the lc-excess values of both mobile and bulk soil water were positively correlated with GWC at 20 and 30 cm depths. When GWC increased due to precipitation, the lc-excess values of mobile and bulk soil water increased. Similarly, when GWC decreased due to evaporation, their lc-excess values also decreased. The lc-excess values of mobile and bulk soil water at all measured depths consistently differed significantly, although GWC varied greatly, suggesting a clear separation between mobile and bulk soil water that is not affected by GWC. This result is consistent with the finding by Evaristo et al. (2016) that ecohydrological separation was consistently present in two tropical catchments with contrasting moisture conditions (Luquillo and Susua catchments in Puerto Rico, with mean annual precipitations of 3700 and 1200 mm, respectively).

A key question is why mobile water separates from bulk soil water isotopically? Some studies indicate that mobile and bulk soil water might be derived from precipitation falling at different times (e.g., Gierke et al., 2016; Allen et al., 2019). In some environments, small pores that contain tightly bound water are preferentially filled by snowmelt from winter and spring, whereas mobile water from summer thunderstorms infiltrates quickly through macropores along preferential flow paths. Due to the seasonal variation in precipitation, winter and summer precipitation have different isotope signals (Bowen et al., 2019), resulting in distinct differences in isotopic patterns between mobile and bulk soil water. Precipitation in winter (December-February) and summer (June-September) accounted for 2% and 77% of total average annual precipitation (464 mm) from 2003 to 2019, respectively, at our study site. Such small amounts of winter precipitation might not be able to fill the small pores. However, our finding that mobile and bulk soil water maintained distinct isotopic signals indicates that this separation may be caused by other factors, and not necessarily by seasonal variation in precipitation.

We also found the degree of separation between the lc-excess of mobile and bulk soil water gradually decreased as the soil depth increased (e.g., 100 cm and 150 cm), for the following reasons. The effect of soil evaporation on bulk soil water gradually weakens with increases in soil depths. Thus, the enriched isotopic signals formed by evaporation in bulk soil water gradually decline or even disappear. In addition, mobile water in deep layers is more likely to be recharged by both preferential and matrix flows than by preferential flow alone (Xiang et al., 2019). Under matrix flow conditions, newly infiltrated water displaces existing 'old water', pushing it deeper into the soil profile and eventually into groundwater (Zheng et al., 2019), so mobile water mixes with tightly bound water. Although the mixing of mobile and tightly bound water conflicts with the original hypothesis of Brooks et al. (2010), evidence of this phenomenon has been provided by Vargas et al. (2017), who found that 75 to 95% of tightly bound water isotopically exchanged with mobile water in a glasshouse experiment with potted *Persea americana* in two contrasting soil types. In addition, Adams et al. (2020) found that mobile and bound soil water isotope ratios are affected by soil texture and mineralogy (e.g., smectite and clay contents). The extent to which tightly bound water mixes with mobile water is unclear at our study site, but such exchange might be one of the reasons for the weakening of the separation



between mobile and bulk soil water in deep layers. These findings suggest that mobile and bulk soil water at all measured depths were continuously separated in the soil matrix, which might be related more to fundamental processes that drive isotopic changes (e.g., soil evaporation and water flow paths) than to soil water conditions. This result is consistent with findings by

Evaristo et al. (2016) and Dubbert et al. (2019) that the isotopic distinction in the TWW hypothesis may be driven by spatiotemporal dynamics of soil water profiles associated with soil evaporation.

### 4.1.2 Isotopic offset between bulk soil water and root water

The root water at 0-60 cm depths showed enriched isotopic signals ($\delta^2H$ and $\delta^{18}O$ values) that were consistent with bulk soil water isotopic signals at the same depth and distinct from mobile water isotopic signals (Fig. 2 and 5). In contrast, at 80-

160 cm depths, $\delta^2H$ and $\delta^{18}O$ values of root water deviated significantly from those of bulk soil water and mobile water, especially $\delta^2H$ values. These results showed that the isotopic offset between plant root water and soil water occurred at the root-soil interface (Fig. 8).

**Fig. 8**

An alternative explanation for isotopic mismatch is that plant rely predominantly on immobile tightly bound water in

accordance with the TWW hypothesis (Brooks et al., 2010; McDonnell, 2014). The rapidity of mobile water's passage through soil reduces its contact with mineral surfaces, and hence nutrient concentrations of mobile water (McDonnell, 2017; Sprenger et al., 2019). Thus, plants preferentially use tightly bound water that is strongly affected by evaporative effects, showing similar enriched isotopic signals and resulting in isotopic separation from mobile water and groundwater. This hypothesis is corroborated by the overlap in isotopic composition between root and bulk soil water at 0-60 cm depths (Fig. 2 and 5). However,

our results showed that bulk soil water did not match root water isotopes at soil depths greater than 80 cm. We considered whether bulk soil water isotopes can represent isotopic values of tightly bound water used by plants. Generally, the water designated 'bulk soil water' includes mobile and tightly bound water due to limitations of water extraction technology when assessing the TWW hypothesis. Thus, the proportion of mobile water in the bulk soil water increases as soil moisture increases, resulting in isotopic deviation between root water and bulk soil water. As shown in Fig. S4, root water does not match tightly

bound water isotopes well at same depth. These results suggest that other factors may also contribute to isotopic offsets in addition to ecohydrological separation.

Another possible explanation is that isotope fractionation occurs during uptake by roots, as previously reported for halophytic and xerophytic plant species (Lin and Sternberg, 1993; Ellsworth and Williams, 2007; Barbeta et al., 2019). Isotopic fractionation can explain isotopic mismatches between root water and bulk soil water at 80-160 cm depths. We speculate that

fractionation also occurs at 0-60 cm depths, but bulk soil water in this layer is strongly affected by evaporation, resulting in the same enriched isotope signals, so isotopically enriched signals caused by soil evaporation mask plant fractionation signals. In contrast, soil evaporation is weaker in deep layers (80-160 cm), leading to a dampened isotopic enrichment signal in bulk soil water, while, root water isotopes are still relatively enriched due to the fractionation effect. Therefore, the isotopic offset between root water and bulk soil water was only observed in the deep soil layer. The maximum differences between bulk soil





water and root water were $-8.6$ and $-1.8‰$ for $\Delta^2H$ and $\Delta^{18}O$, respectively: lower than the differences between bulk soil water and xylem water reported by Barbeta et al. (2020) (ca. $-11$ and $-2‰$ for $\Delta^2H$ and $\Delta^{18}O$, respectively) and Poca et al. (2019) ($-24.6$ and $-2.9‰$ for $\Delta^2H$ and $\Delta^{18}O$, respectively). The inconsistency in results may be related to variations in arbuscular mycorrhiza (Poca et al., 2019), soil water loss and soil type (Vargas et al., 2017), fractionation in root-xylem water transport (Martin-Gomez et al., 2017) and plant species (Dubbert et al., 2019). We also detected a positive linear relationship between

$\Delta^{18}O$ and $\Delta^2H$ in both soil-root and soil-xylem offsets (Fig. 6), suggesting that both hydrogen and oxygen isotopes changed simultaneously, in accordance with findings by Vargas et al. (2017) and Barbeta et al. (2020). Overall, these findings suggest that the isotopic offset between soil water and root water is likely governed by ecohydrological separation and plant fractionation, and may be not due solely to either of them, supporting our second hypothesis.

**4.2 Root water and xylem water isotopes**

We found that xylem water mainly overlapped isotopically with root water at 100-160 cm depths (Fig. 2), while the isotopically enriched root water at 0-80 cm depths was not reflected in the xylem water isotopes. Under the assumption that plant fractionation does not occur, one possibility is that trees preferentially use more deeper soil water and groundwater than fluctuating shallow soil water, which is a less stable and reliable water source because it is subject to rapid evaporation and seasonal precipitation (Zhao and Wang, 2018). Furthermore, *S. matsudana*'s deep water use strategy may provide favorable

water conditions for shallow-rooted herbaceous species, facilitating stable coexistence. Roots at 0-80 cm depths absorb less water with enriched isotope signals than deep roots. A small proportion of the isotopically enriched root water fully mixes with isotopically depleted root water in deep layers, resulting in the disappearance of isotopically enriched signals in the xylem water.

    Another possibility is that water isotope heterogeneity in plant tissues may contribute to the isotopic deviation. Xylem

water might not reflect current root water isotopic signals because it takes time to transport water from roots to branches and water also has residence times in branches and roots (Penna et al., 2018; Allen et al., 2019). For example, using deuterated water ($D_2O$) and heat as tracers to analyze water transport from the base of the trunk to the upper crown, Meinzer et al. (2016) found that transit times ranged from 2.5 to 21 days and residence times ranged from 36 to 79 days. However, we found that $\delta^2H$ and $\delta^{18}O$ values of xylem water remained stable (mean values: $-66.68 \pm 1.61$ and $-7.71 \pm 0.24‰$, respectively) during

our high frequency (ca. 3-day) sampling period from August 4 to September 15, 2019 (Fig. 3), which does not support this interpretation. Moreover, to test the possibility that isotope enrichment may have been present in the unsampled branches we collected xylem water at different tree heights (150-450 cm) on August 18, 2019, and found no significant differences ($p > 0.05$) (Fig. S5). In addition, previous studies have also provided indications that xylem water isotope was more enriched than that of potential water sources due to fractionation effect during water transport in the xylem. This phenomenon has been

reportedly associated with temporal declines in sap flow rates (Martin-Gomez et al., 2017), water exchange between phloem and xylem (Cernusak et al., 2005) and leafless or newly leafed deciduous species (e.g., *Quercus laevis* and *Carya floridana*) (Ellsworth and Sternberg, 2015). However, we found that the xylem water contained more unenriched isotopic signal from





deep roots than enriched isotopic signal from shallow roots. The result show that there was no fractionation during water transport from root to xylem (Fig. 8). Thus, we used root water isotopes to quantify the proportional use of soil water at

different depths, and found that water from 100-160 cm layers accounted for 74% of the total (Fig. 7). In conclusion, replacement of soil water by root water in analyses provides a means for accurately quantifying plant water sources.

## 5 Conclusion

At our study site during the covered period, xylem water of *S. matsudana* trees was isotopically isolated from bulk soil water, mobile water and groundwater, supporting our first hypothesis that isotopic offset occurred between xylem water and

potential water sources. We further detected the cause and location of this mismatch. In the soil matrix, bulk soil water generally had higher $\delta^2H$ and $\delta^{18}O$ values than mobile water, due to effects of soil evaporation and water flow paths, following the isotopic patterns in the TWW hypothesis. The isotopic composition of root water overlapped with that of bulk soil water at 0-60 cm depths, in association with plants' use of bulk soil water as in the TWW framework. However, root water deviated significantly from bulk soil water isotopically at 80-160 cm depths and the maximum difference between bulk soil water and

root water was −8.6 and −1.8‰ for $\delta^2H$ and $\delta^{18}O$, respectively. These findings suggest isotopic offset occurred at root-soil interface, probably due to a combination of ecohydrological separation, as in the TWW hypothesis, and plant fractionation, supporting our second hypothesis. In contrast, no isotopic fractionation occurred during root to xylem water transport. Isotopically, xylem water of *S. matsudana* trees mainly overlapped with root water isotopes at 100-160 cm depths and the contribution of these root layers to xylem water reached 74%. Our results challenge current understanding of the behavior of

H and O isotopes, extending our knowledge of isotopic signals in the soil-root-xylem continuum and providing valuable insights into fundamental ecohydrological process.

## Data availability

The data that support the findings of this study are available from the corresponding author upon request.


## Author contributions

LW conceptualized this research. YZ collected the data. Both authors contributed to the writing of the manuscript.

## Conflicts of Interest

The authors have no conflicts of interest to declare.

## Acknowledgments





This work was supported by the National Natural Science Foundation of China (grant nos. 41771545 and 41977012), the Strategic Priority Research Program of Chinese Academy of Sciences (XDB40000000), and the State Key Laboratory of Loess and Quaternary Geology, Institute of Earth Environment, CAS (ref no. SKLLQG1718).

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

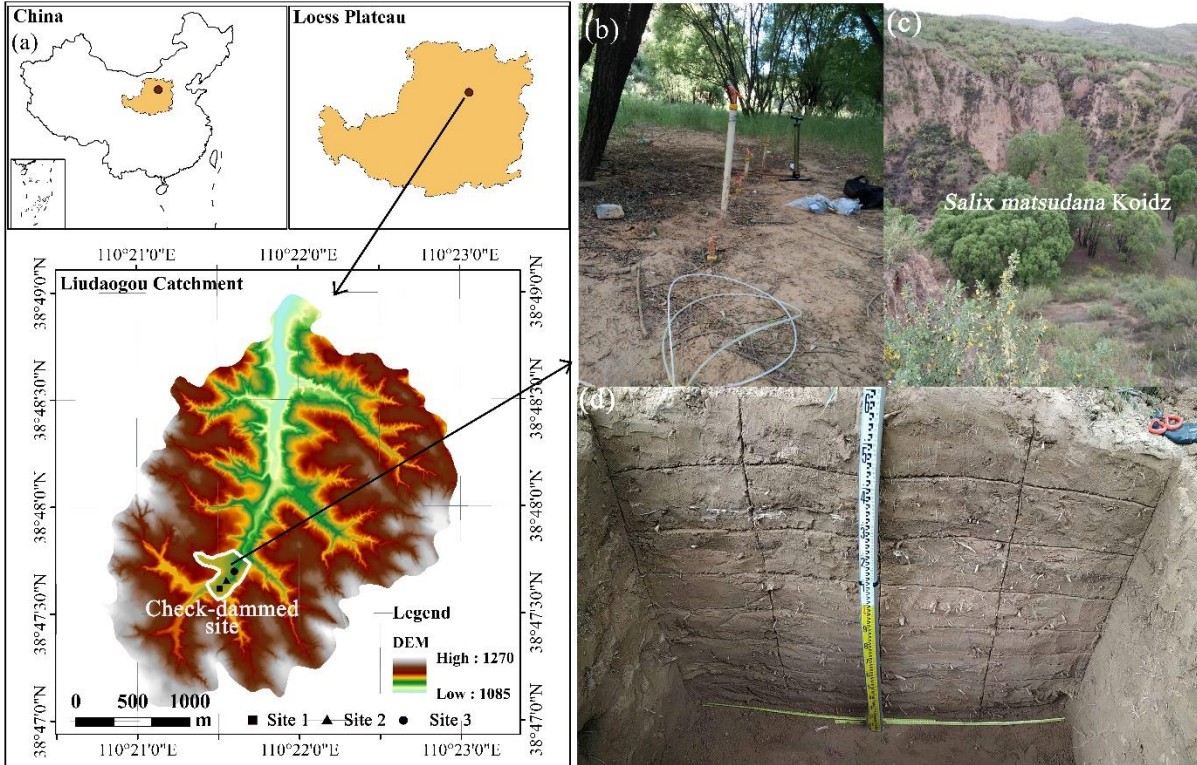

**Figure 1: (a) Location of the study area on the Loess Plateau, China. (b) Photograph of mobile water collection using suction lysimeters (white plastic tubes) (with application of 60 kPa tension), and (c) *Salix matsudana* Koidz, our sampling tree. (d) Profile of the soil cuboid (length, width and depth: 160, 80 and 160 cm, respectively) being dug to obtain root isotopic data. The soil cuboid was divided into 64 sub-cuboids and root isotope in each sub-cuboid (length, 40 cm; width, 40 cm; height, 20 cm) were collected separately.**




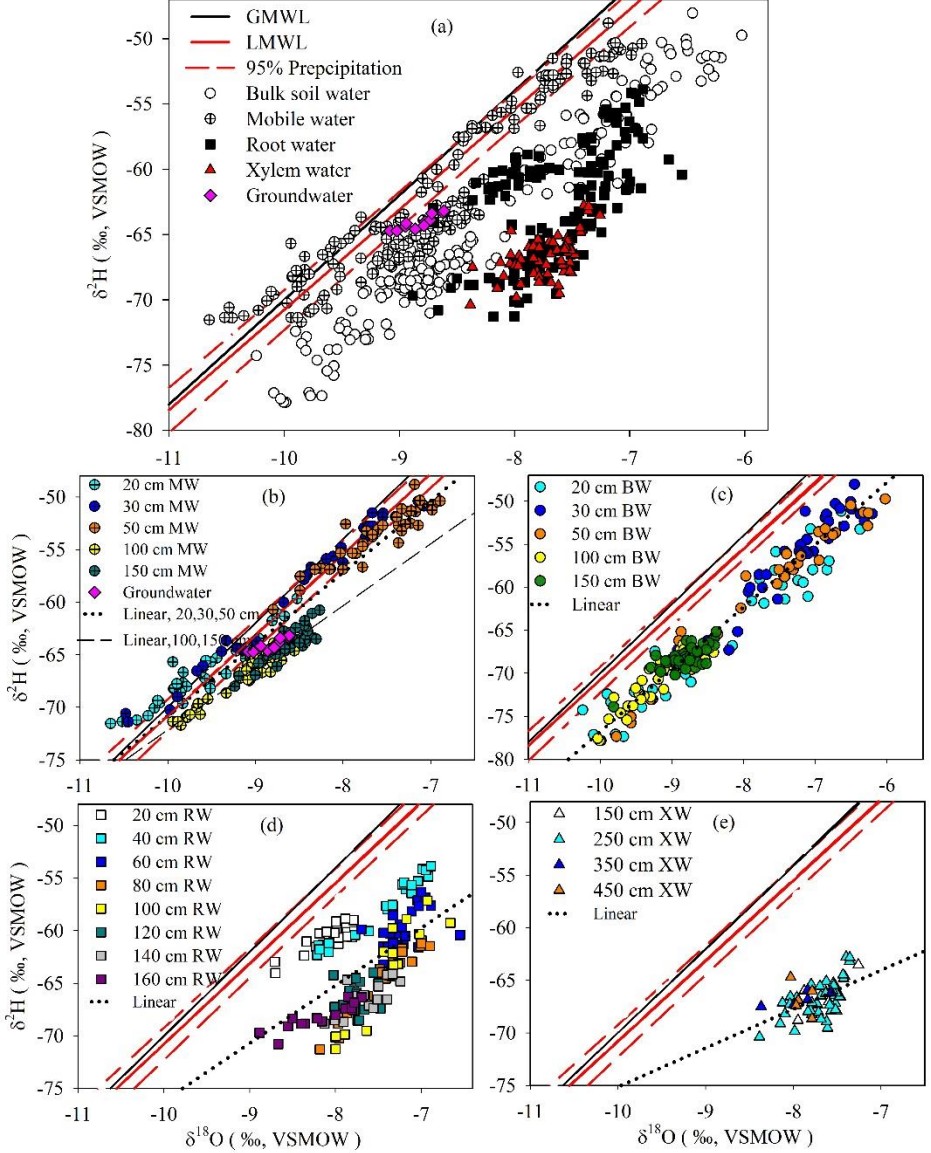

**Figure 2 (a) δ¹⁸O and δ²H isotope values collected from August 4 to September 15, 2019. Plotted values include bulk soil water (BW), mobile water (MW), root water (RW), xylem water (XW) and groundwater. (b) δ¹⁸O and δ²H isotope values of groundwater, and MW collected from different depths, (c) BW collected from different depths, (d) RW collected from different depths, and (e) XW collected from different tree heights. The red line represents the 2016-2019 local meteoric water line (LMWL, δ²H = 5.91 + 7.67 δ¹⁸O, R² = 0.96) and 95% confidence interval of precipitation. The black line represents the global meteoric water line (GMWL, δ²H = 10 + 8 δ¹⁸O). The dotted black lines represent the linear regressions.**



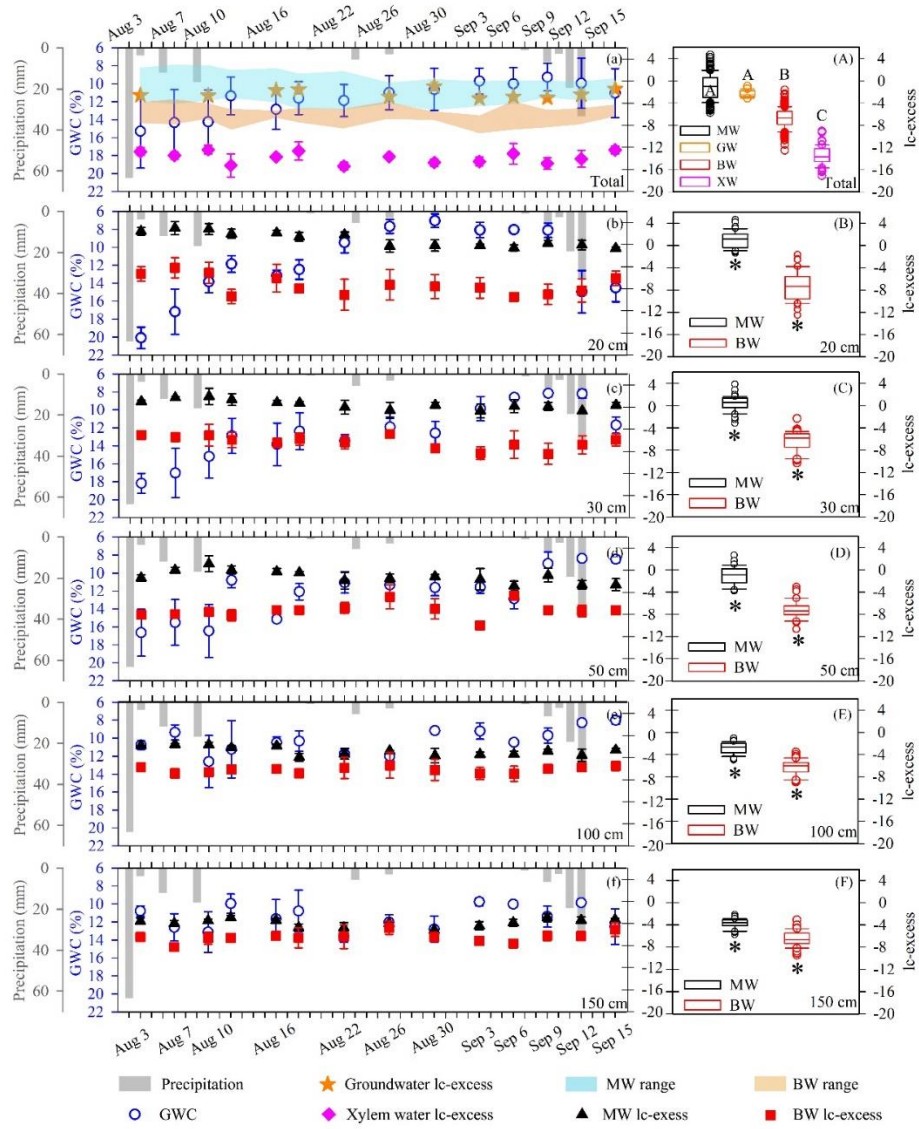

**Figure 3 (a-f) Temporal dynamics of hydrological conditions (precipitation and gravimetric water content, GWC) and lc-excess values (these values are means and standard deviations for three sites) of groundwater (GW), xylem water (XW), mobile water (MW) and bulk soil water (BW) at indicated depths (20, 30, 50, 100 and 150 cm) during the period August 3 to September 15, 2019. (A) Boxplots of total MW (N=191), GW (N=22), BW (N=204) and XW (N=61) lc-excess values. (B-F) Boxplots of MW and BW at 20 cm (MW, N=40; BW, N=42), 30 cm (MW, N=40; BW, N=40), 50 cm (MW, N=38; BW, N=40), 100 cm (MW, N=36; BW, N=40) and 150 cm (MW, N=37; BW, N=42) depths. The top and bottom of each box are the 25th and 75th percentiles of the samples, respectively. The black line in each box is the sample median. Xylem water and potential water sources that do not share a letter are significantly different (p < 0.05, Tukey-Kramer HSD). Asterisks show significantly differing lc-excess values between mobile water and bulk soil water at the same depth (p < 0.05).**





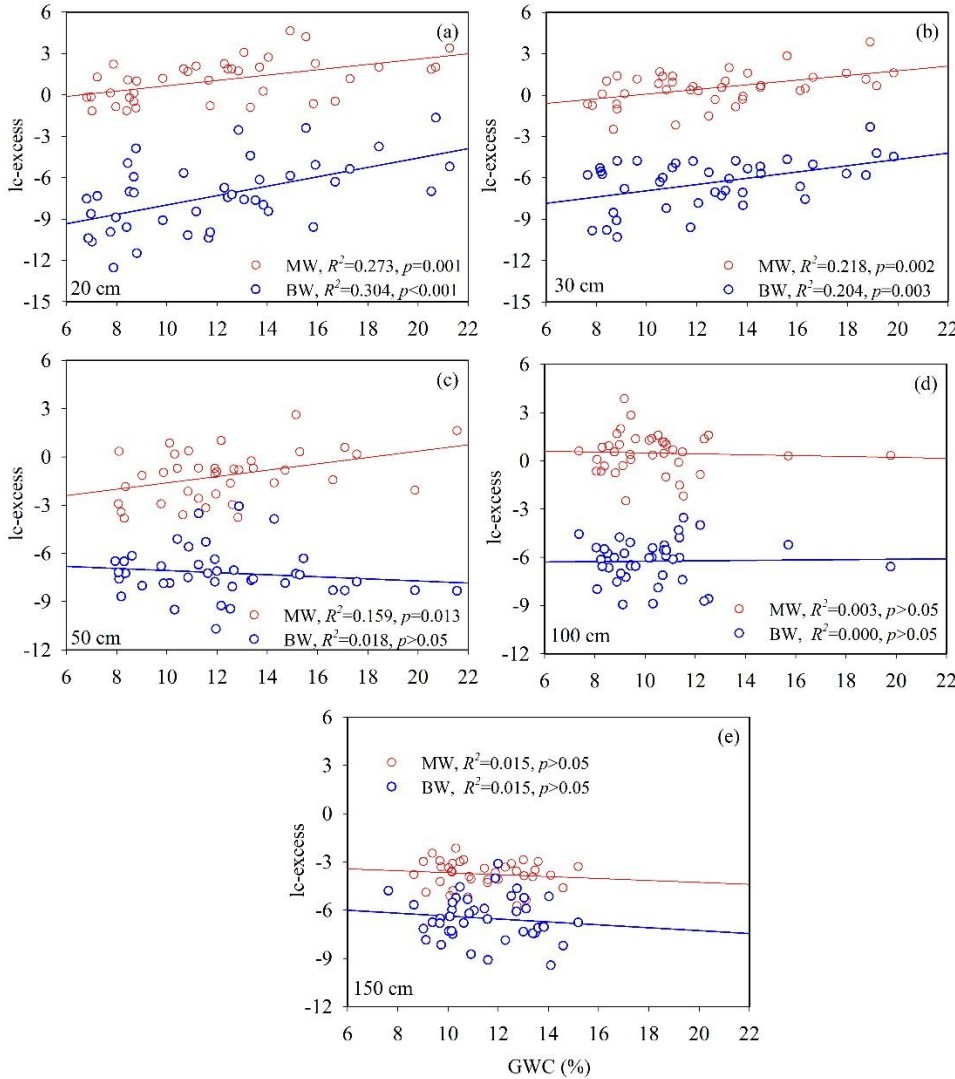

**Figure 4 Relationships between gravimetric water content (GWC) and (a) lc-excess values at 20 cm depth, (b) lc-excess values at 30 cm depth, (c) lc-excess values at 50 cm depth, (d) lc-excess values at 100 cm depth and (e) lc-excess values at 150 cm depth. Data from lc-excess values of mobile water (MW) and bulk soil water (BW) are shown in red and blue circles, respectively. The insets show the fitness of the linear regressions.**

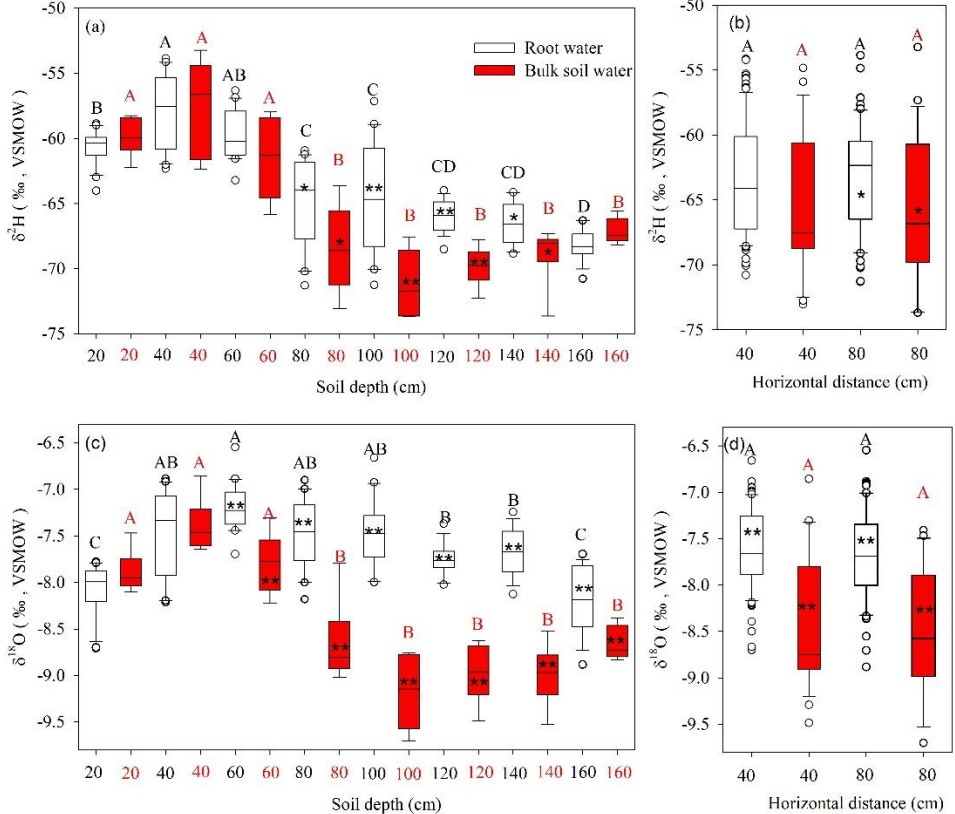

**Figure 5 Boxplots of root water and bulk soil water isotope composition ($\delta^2H$ and $\delta^{18}O$) at indicated depths (a, c) and horizontal distances from the tap root of the focal root system (b, d). The top and bottom of each box are the 25th and 75th percentiles of the samples, respectively. The black line in each box is the sample median. Asterisks indicate significantly differing isotopic values between soil water and root water (\* and \*\*: $p < 0.05$ and $p < 0.01$, respectively, according to two-tailed tests). Plant root water isotopes or bulk soil water isotopes at different depths that do not share a letter are significantly different ($p < 0.05$, Tukey-Kramer HSD).**





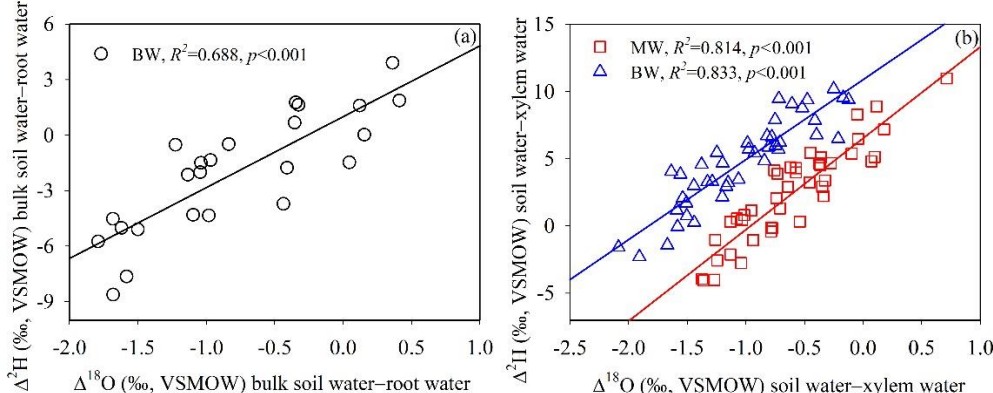

**Figure 6 (a) Relationship of hydrogen isotope offset ($\Delta^2H$, $\Delta^2H = \delta^2H_{soil} - \delta^2H_{root}$) and oxygen isotope offset ($\Delta^{18}O$, $\Delta^{18}O = \delta^{18}O_{soil} -$**

535 **$\delta^2O_{root}$) between bulk soil water and root water, according to analyses of samples of bulk soil water (BW) and root water collected from 0-160 cm depths on August 18, 2019. (b) Relationship of hydrogen isotope offset ($\Delta^2H$, $\Delta^2H = \delta^2H_{soil} - \delta^2H_{xylem}$) and oxygen isotope offset ($\Delta^{18}O$, $\Delta^{18}O = \delta^{18}O_{soil} - \delta^2O_{xylem}$) between soil water and xylem water, according to analyses of samples for bulk soil water, mobile water (MW) and xylem water collected from August 4 to September 15, 2019. The insets show the fitness of the linear regressions (a-b).**

540





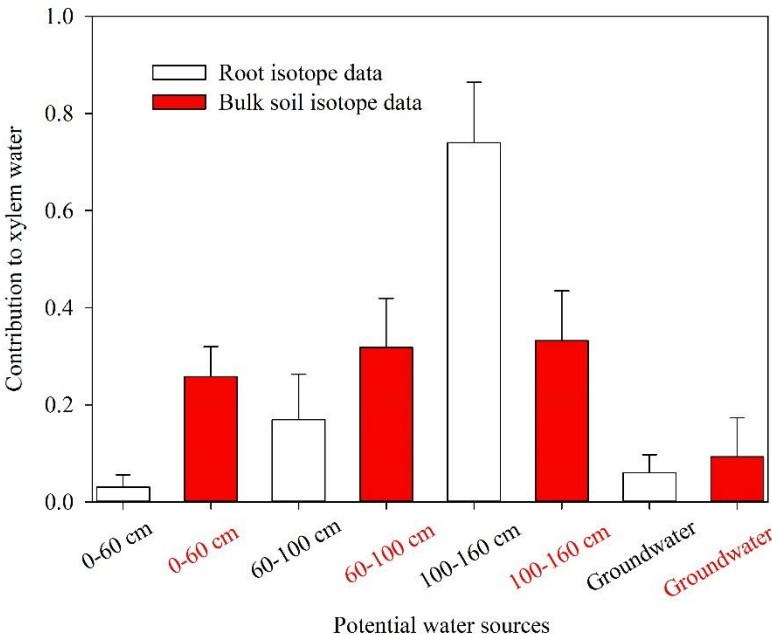

545 **Figure 7 Contributions of potential water sources to plant xylem water, based on analyses of samples of root water and bulk soil water isotopes collected on August 18, 2019.**





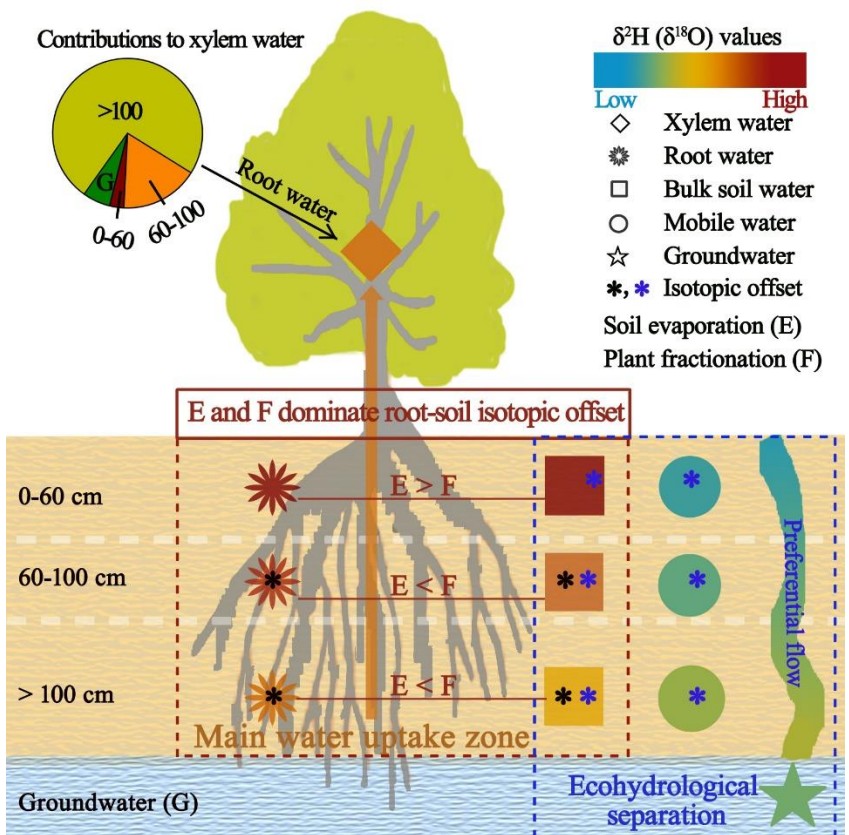

**Figure 8 Schematic diagram of isotopic dynamics along the soil-root-xylem continuum. Color codes indicate isotopic signals of mobile water, bulk soil water and root water at indicated depths, groundwater and xylem water (from blue to brown representing low to high). The pie chart represents the contributions of potential water sources to xylem water obtained using the Bayesian mixing model SIAR based on both $\delta^2H$ and $\delta^{18}O$ values of root and groundwater (G) samples. The black asterisks indicate significant differences in the isotopic offset between root water and bulk soil water at the same depth ($p < 0.05$). The blue asterisks indicate significant differences in the isotopic offset between mobile water and bulk soil water at the same depth ($p < 0.05$).**





**565** **Table 1 Distribution of soil particle composition according to the USDA soil texture classification system**

| Soil depth (cm) | Soil particle composition (%) | | | Soil texture |
|---|---|---|---|---|
| | Sand | Silt | Clay | |
| 10 | 56.76 | 34.78 | 8.46 | Sandy loam |
| 20 | 64.45 | 28.31 | 7.24 | Sandy loam |
| 30 | 67.65 | 25.69 | 6.66 | Sandy loam |
| 40 | 53.20 | 37.96 | 8.84 | Sandy loam |
| 50 | 60.67 | 31.18 | 8.15 | Sandy loam |
| 60 | 39.07 | 48.44 | 12.50 | Loam |
| 70 | 60.54 | 31.22 | 8.24 | Sandy loam |
| 80 | 50.55 | 40.38 | 9.07 | Loam |
| 90 | 51.06 | 39.06 | 9.88 | Loam |
| 100 | 61.05 | 30.43 | 8.52 | Sandy loam |
| 120 | 63.81 | 29.38 | 6.82 | Sandy loam |
| 140 | 51.31 | 39.17 | 9.52 | Loam |
| 160 | 45.29 | 44.09 | 10.61 | Loam |