# Peer review of "Insights into isotopic mismatch between bulk soil water and *Salix* matsudana Koidz trunk water from root water stable isotope measurements"

_Hydrology and Earth System Sciences, 2020_

## Author Comment (AC1)

**Response to Reviewer 1, re: "Insights into isotopic mismatch between soil water and *Salix matsudana* Koidz xylem water from root water isotope measurements", in review in HESSD (NO. hess-2020-680).**

We thank Reviewer #1 for thoughtfully and critically reviewing our manuscript. We greatly appreciate the positive feedback and many well-founded points that have certainly helped us to improve the manuscript. Overall, we agree with these suggestions and have made targeted amendments, as described in the detailed point-by-point replies to the Reviewer' comments below. The reviewer's comments are presented in blue, and passages changed in specific responses to the comments are presented in quotation marks and italic font.

**Point-by-point by responses to Reviewer 1's comments**

*Major points:*

1. The manuscript hess-2020-680: "Insights into isotopic mismatch between soil water and *Salix matsudana* Koidz xylem water from root water isotope measurements" by Zhao and Wang investigates potential reasons for an observed mismatch between soil and xylem water stable isotope values by measuring water stable isotope ratios in different soil water pools (mobile, bulk soil and bound) across soil depths, as well as in roots (across depth) and xylem of three rigorously sampled tree individuals. The authors conclude that the observed isotopic differences between xylem and bulk soil water arise from a combination of ecohydrological separation, i.e. isotopic differences in mobile and bound soil water and plant fractionation during root water uptake.

Root water uptake depth is routinely determined by comparing the isotope composition of xylem water with that of soil water in different depths (and other water sources like stream and groundwater) assuming that extracted water from bulk soil samples represents available water sources. This long-standing principle is lately repeatedly questioned and a lot of uncertainty persists on potential reasons and underlying causes. This work contributes to the discussion and provides new insights. I especially liked, that the authors conducted an experiment under natural conditions. I do not know of any other study that sampled water sources, xylem water and also systematically investigated root xylem within a field experiment to tackle this question and I applaud the authors for conducting this surely very labour-intensive work. The manuscript is well-structured and understandable. Generally, I am in favour of publishing this work. However, I think that a number of critical points (see major points below) should be addressed beforehand in a revised version. I also suggest an English native speaker to proof-read the manuscript and help to further improve some of the expressions.

**Reply:** We thank the reviewer for the constructive comments and suggestions, we have carefully considered them and tried our best to address the highlighted weaknesses in the manuscript. In addition, we will invite a professional native English-speaking editor with a PhD in a relevant discipline to edit the next version.

2. You state that plant fractionation, i.e. change of isotope values during root water uptake, is one of two main causes for the observed mismatch. This was observed before in xerophytic and halophytic plants and for plants in symbiosis with arbuscular mycorrhiza (e.g. Ellsworth & Williams 2007, Poca et al. 2019). However, previous studies on plant fractionation reported depleted (more negative) isotope values in plant xylem as compared to soil water,

hence plants discriminated against the heavy isotope (mostly $^2$H). This is not in line with

enriched (less negative) xylem values reported here. While it cannot be ruled out completely

that water would get enriched in heavy isotopes during rwu, this was not reported before and

considering other potential reasons, namely isotopic heterogeneities across soil water pools as

well as temporal variability and methodological artefacts, this seems unlikely to me. If you

decide to keep it in the manuscript, this discrepancy to other studies should be pointed out

and discussed in detail.

**Reply:** Thanks for your suggestions. We would like to change the conclusion that isotopic

fractionation leads to the observed mismatch between root water and bulk soil water at the same

depth in the manuscript, for two reasons. First, recent studies on isotopic fractionation have found

stronger $^2$H depletion in trunk water/root water than in bulk soil water (e.g., Poca et al., 2019;

Vargas et al., 2017). However, these findings are not consistent with our finding that root water

had higher $\delta^2$H values than bulk soil water (up to 8.6‰) as suggested. Second, as pointed out by

Reviewer #2, the water in the sampled coarse roots (> 2 mm diameter) does not necessarily match

the bulk soil water around them because sampled coarse roots can transport and mix water from

different locations. Most importantly, we found that the isotopic composition of root water

deviated from that of bulk soil water, but overlapped with the values derived for less mobile water

(see Figure 1 below). Thus, we concluded that soil-root isotopic offsets are more likely to be

caused by the complexity of root systems and the heterogeneity of bulk soil water than isotopic

fractionation during root water uptake. Hence, we would like to add the following discussion

regarding this issue in the next version:

[revised manuscript text omitted]

2017.

3. Building on that, some root and all stem xylem samples show an evaporative enrichment in

the dual isotope plot (Fig. 2). This is discussed in the manuscript and related to an enriched

signal of bound soil water. I think this argument would be strengthened if you provide further

description on the sampling procedure. Specifically, I wondered if evaporation during root

sample collection could potentially influence obtained results. Did you sample roots right

from the soil profile wall or exclude the first few centimetres? How fast was sampling

conducted after digging the hole? Also: Was soil thoroughly removed from sampled roots?

Regarding xylem sampling, you write "Bark was peeled from the twigs and all leaves were

removed to avoid perturbance of xylem water isotopic signatures by fractionation." (L 136-

137). So, did you sample twigs that had leaves directly attached to them? Could the enriched

signal hence arise from back-diffusion or any other exchange with enriched leave water before sampling? Were twigs fully suberized or were they green and hence photosynthetically active?

**Reply:** We would like to add details about the root and trunk water (changing xylem water to trunk water, as suggested) sampling and schematic diagram of root excavations (Figure 2) in the *Materials and methods* section, as follows:

Regarding root sampling:

*"We excavated a soil cuboid with 160 cm depth, 80 cm width (horizontal distance) and 160 cm length with the main root of the selected tree at the center (Fig. 2a). We then divided the cuboid into 64 sub-cuboids (length, 40 cm; width, 40 cm; height, 20 cm) (Fig. 2b) and dug each sub-cuboid one by one to minimize risks of evaporation. 2-3 coarse roots (> 2 mm diameter) from each sub-cuboid were randomly selected and roots from the top few centimeters of the topsoil were not artificially removed. To minimize the influence of attached soil on root water, these sampled roots were rapidly peeled to remove bark, placed in 10 mL vials and sealed with caps then the caps were secured with Parafilm. Finally, these samples were kept in a cool box until storage in the lab at 4℃. To compare the isotopic composition of root and bulk soil water at the same depths, we collected samples of soil around the sampled roots in each sub-cuboid. These soil samples were also rapidly placed in 10 mL vials that were sealed in the same manner as the root samples, then kept in a cool box until storage in the lab at −20 ℃."*

Regarding trunk sampling

*"Tree samples were collected simultaneously with the soil samples. These consisted of twigs collected from the south-facing side of three* S. matsudana *trees at 250 cm height on each*

*sampling occasion. In addition, samples of trunk at selected tree heights (150, 250, 350, 450 cm) were collected on August 18, 2019. Bark and phloem were peeled from fully suberized branches to avoid perturbance of trunk water isotopic composition by fractionation. Pieces of the de-barked and de-leaved twigs, 30 mm long, were then immediately placed in 10 mL vials, the vials were sealed with caps then the caps were secured with Parafilm. These samples were also kept in a cool box until storage in the lab at 4°C."*

[Figure]

Figure 2: Schematic diagram of root excavations (a) and measurements (b).

4. In my personal opinion, I would not put so much emphasise on the TWW hypothesis. If a reader is not familiar with it, it might be a bit confusing (especially in the abstract). I think it is good that you refer your results to it but I would reduce the importance it has in your manuscript, e.g the amounts of mentions.

Also, I think the term "tightly bound water" is misleading. At least in my perception it suggests that plants use an exceptionally tightly bound water pool (as compared simply to bound water). However, in my opinion, they probably just use the water that is available to them and mobile water might infiltrate beyond the root zone too quickly to be available in the

long term. I imagine that the sampled mobile water (in lysimeters) mainly originates from percolation during precipitation events, when relative humidity is high, solar radiation is low and plants hence do not transpire a lot. I suggest to use "less mobile water" or simply "bound water". However, I am aware that it is also termed "tightly bound water" in the TWW hypothesis.

**Reply:** We agree that references to the TWW hypothesis should be reduced. Moreover, the short experimental period and the focus on phenomenon that are not directly related to the TWW hypothesis hinder concise, meaningful discussion of the hypothesis and ecohydrological separation, as pointed out by Reviewer #2. Therefore, we would like to delete all content about the TWW hypothesis and pay more attention to the soil water's heterogeneity through the comparison of mobile water, bulk soil water, and derived characteristics of less mobile water (we intend to change 'tightly bound water' to 'less mobile water' as suggested) at the same depths, and the impact of this heterogeneity on plant water uptake in the next version.

5. You took the time to calculate the isotopic composition of tightly bound soil water (from measured bulk and mobile soil water) and you repeatedly argue that plants preferentially use this water source. However, this data is only incorporated into one supplemental figure. I think it would strengthen the story if those values were incorporated into the main figures as well (e.g. Figure 2 and 5).

**Reply:** We plan to add data on the less mobile water stable isotopes in the supplemental figures to the main figures as suggested. Please see Figure 1 above and Figure 3 below.

[Figure]

Figure 3 (a-f) Temporal dynamics of hydrological conditions (precipitation and gravimetric water content, GWC) and lc-excess values (these values are means and standard deviations for three sites) of groundwater (GW), trunk water (TW), mobile water (MW), less mobile water (LMW) and bulk soil water (BW) at indicated depths (20, 30, 50, 100 and 150 cm) during the period August 3 to September 15, 2019. (A) Boxplots of total MW (N=191), GW (N=22), BW (N=204), TW (N=61) and LMW (N=176) lc-excess values. (B-F) Boxplots of MW and BW at 20 cm (MW,

N=40; BW, N=42; LMW, N=39), 30 cm (MW, N=40; BW, N=40; LMW, N=34), 50 cm (MW, N=38; BW, N=40; LMW, N=33), 100 cm (MW, N=36; BW, N=40; LMW, N=34) and 150 cm (MW, N=37; BW, N=42; LMW, N=36) depths. The top and bottom of each box are the 25th and 75th percentiles of the samples, respectively. The black line in each box is the sample median. Trunk water and potential water sources that do not share a letter are significantly different (p < 0.05, Tukey-Kramer HSD).

6. SIAR modelling: In my opinion, this does not strengthen the story. The isotopic composition of water sources incorporated are quite similar to each other. Did you check if the modelling results change substantially when running the calculations multiple times? Also, you use bulk soil water in different soil depths as available water sources. Contradictorily, these do not match with observed xylem values and you argue in the text that plants use soil water with differing isotopic composition. Additionally, measurement uncertainty in sources and plant xylem should be considered (see e.g. Kühnhammer et al. 2020). Having that said, I like that you compare rwu fractions derived with bulk soil samples and sampled roots as water sources. If you decide to keep the SIAR modelling, it would be interesting, in my opinion, to also look at tightly bound water as a potential set of sources and see how this changes the results. You could then also discuss the weaknesses of those purely statistical models and make use of your data to communicate potential issues with the usual approach (e.g. comparing bulk soil water and xylem water) to the scientific community.

**Reply:** We would like to delete the calculation of plant water source contributions based on SIAR modeling as suggested, but keep the conclusion that root water at 100-160 cm depths was the main water source for the sampled plants. We believe that root water can reflect the water source of

trees better than bulk soil water (which has been more extensively used), for reasons detailed in our response to comment 2.

7. As you sampled roots across soil depths, do you also have information on root length density across the profile or any other measure of root quantity across soil depths? I think this could add some interesting insights into the trees' water uptake strategies.

**Reply:** Unfortunately, we did not collect other root information such as root length density due to the large workload.

*Technical comments:*

8. As you are discussing isotopic differences of soil water pools sampled as a reason for the observed mismatch to xylem water, you should pay close attention to specifying which soil water pool you are talking about. I suggest, either you always specify this or you clarify once that when talking about soil water in general you always refer to bulk soil water.

**Reply:** Thanks for your suggestions. We will identify the soil water we are referring to in the revised version.

9. In the copy I received for review, the figure quality seems too low, axis labels and names seem a bit blurry. It should be verified that all labels and names are easily readable.

**Reply:** The resolution of the figures was reduced by conversion from Word to pdf format. We plan to address to this issue in the revised manuscript. In addition, we will replot the figures to improve their resolution.

10. Check for consistency of isotope terminology and avoid using (too many) different ones if it does not contribute to the readability of the text. You use all of the following: "isotopic

composition", "isotope composition", "isotopes in water", "isotopic signature", "isotope

signals", "isotopic signals", "isotope fingerprints", "H and O isotopes", "hydrogen and

oxygen isotopes", "water isotopes", "isotopic values", "isotopic patterns"

I know that sometimes it makes sense to mix it up a bit to avoid too many repetitions but I

think you can eliminate at least a few of them :-D

Also I would use the term "water stable isotopes" instead of "water isotopes"

**Reply:** Thanks for your suggestions. We will apply more consistent isotope terminology in the

next version.

11. You use xylem to refer to measurements in the trunk. However, water within roots is also

    transported in the xylem (which you sampled to obtain the root isotopic values). I suggest to

    specify the use of words here or use trunk instead of xylem

**Reply:** We will change xylem water to trunk water as suggested in the next version.

***Specific comments:***

12. Title: "insights into [an] isotopic mismatch between [bulk] soil water and *Salix matsudana*

    Koidz xylem water from root water isotope measurements"

**Reply:** As advised, we intend to revise the title in the next version, as follows:

*"Insights into isotopic mismatch between bulk soil water and* Salix matsudana *Koidz trunk water*

*from root water stable isotope measurements"*

13. Figure 2: I think the axis limits should be the same in all subplots (x-axis different for panel

    c).

**Reply:** We intend to standardize the x- and y-axis scales of the plots (see Figure 1 above) in the next version as suggested.

14. Figure 4 (caption): remove repetition of lc-excess

**Reply:** We will also remove the repetition of lc-excess in the next version as suggested.

15. Figure 7: SD should also be displayed in the opposite direction, are the obtained distributions (of a RWU fraction at a certain depth) normally-distributed? If not, display uncertainty in a different way. Maybe it would make sense to display those distributions as boxplots

**Reply:** As mentioned in our reply to comment 6, we would like to delete Figure 7 due to the removal of references to the SIAR model.

16. Figure 8: I like the colour coding depicting the isotopic composition of the different compartments studied. Maybe this would come out even clearer if tree and soil background colours were a little more subtle (maybe grey scale?). Also see my general comment 1 to revaluate if you should emphasize fractionation as the main cause for the isotopic mismatch

**Reply:** We will replot Figure 8 in the next version, as follows:

[Figure]

Figure 4 Schematic diagram of isotopic dynamics along the soil-root-trunk continuum. Color codes indicate isotopic composition of mobile water, bulk soil water and root water at indicated depths, groundwater and trunk water (from blue to brown representing low to high). The black asterisks indicate significant differences in the isotopic offset between root water and bulk soil water at the same depth ($p < 0.05$). The blue asterisks indicate significant differences in the isotopic offset between mobile water and bulk soil water at the same depth ($p < 0.05$).

17. Figure S2: Symbols are different sizes in different subplots

**Reply:** We will replot Figure S2 and keep the same size for each subplot as suggested.

18. Figure S3: What is displayed? The figure legends and part of the caption states you display bulk water (BW) and tightly bound water (TW), but the caption also mentions mobile water (MW) – I assume that is a typo. I would also suggest changing the colours in Figure S3 to match with the colours in Figure 3, i.e. bulk soil water should always have the same colour

**Reply:** We would like to delete Figure S3 and add the isotopic data in it to Figure 3 (see above).

19. Figure S4: the colours of the boxplots are hard to distinguish

**Reply:** We would like to delete Figure S4 due to changes in the text.

20. Line 12: "at high temporal resolution" it reads as if all before mentioned parameters were sampled over time. However, this is not true for root xylem (only sampled on one occasion). Also, I think the perception of high temporal resolution is quite different depending on who you ask, especially with new in situ methods evolving. Maybe specify that you sampled twice a week/every X days?

**Reply:** We plan to clarify this as follows:

*"Thus, we measured the specific isotopic composition ($\delta^2H$ and $\delta^{18}O$) of each component (e.g., bulk soil water, mobile water, groundwater, trunk water and root water of* Salix matsudana *Koidz trees) with about three-day resolution in the soil-root-trunk continuum."*

21. Line 12-13: "to analyze isotopic dynamics in the soil-root-xylem continuum"

I don't really see a lot of focus on (temporal) isotope dynamics in your manuscript. You do not really discuss the variations over time (and only sampled roots once), maybe apart from the influence of GWC on soil water isotopes

**Reply:** We would like to revise this sentence in the next version, as follows:

*"Thus, we measured the specific isotopic composition ($\delta^2H$ and $\delta^{18}O$) of each component (e.g., bulk soil water, mobile water, groundwater, trunk water and root water of* Salix matsudana *Koidz trees) with about three-day resolution in the soil-root-trunk continuum."*

22. Line 17: I personally would not mention the TWW here again. "and plant fractionation" see my major point 1)

**Reply:** We would like to rephrase the Abstract in the next version because of the changes in conclusions such as those regarding the TWW hypothesis (see our response to major point 4) and plant fractionation (see our response to major point 2), as follows:

*"Increasing numbers of field studies have detected isotopic mismatches between plant trunk water and its potential sources. However, the cause of these isotopic offsets is not clear and it is uncertain whether they occur during root water uptake or during water transmission from root to trunk. Thus, we measured the specific isotopic composition ($\delta^2H$ and $\delta^{18}O$) of each component (e.g., bulk soil water, mobile water, groundwater, trunk water and root water of* Salix matsudana Koidz *trees) with about three-day resolution in the soil-root-trunk continuum. We report three main findings. First, we detected clear separation between mobile water and bulk soil water isotopic composition, but the distinction between mobile water and bulk soil water gradually decreased with increasing soil depth. Second, root water deviated from bulk soil water isotopic composition, but it overlapped with the composition derived for less mobile water. The maximum differences in $\delta^2H$ and $\delta^{18}O$ between bulk soil water and root water were −8.6 and −1.8‰, respectively. Third, trunk water was only isotopically similar to root water at 100-160 cm depths, and it remained stable during the experimental period, suggesting that the trees consistently used the stable deep water source. In conclusion, the isotopic offset between bulk soil water and trunk water of* S. matsudana *reflected an isotopic mismatch between root water and bulk soil water associated with heterogeneity of the soil water. Our results illuminate relationships between the*

*isotopic composition of soil water of various mobility, root water and trunk water that may be useful for advancing our understanding and representation of root water uptake and transport."*

23. Line 20: "isotopic offset occurred at the interface between the soil and *S. matsudana* roots"

    This statement is a bit misleading I think because if different soil water pools were not isotopically well mixed, the offset does not occurred at the interface between soil and roots but because plants only access a certain soil water pool

**Reply:** We will amend the sentence as follows:

*"The isotopic offset between bulk soil water and trunk water of* S. matsudana *reflected an isotopic mismatch between root water and bulk soil water associated with heterogeneity of the soil water."*

24. Line 28: "in [the] global hydrological cycle" or "in hydrological cycles", "terrestrial ecosystem[s]"

**Reply:** We plan to amend this sentence, as follows:

*"RWU also controls partitioning of infiltrated soil water between groundwater recharge and local atmospheric return through evapotranspiration (Knighton et al., 2020a; Knighton et al., 2020b), and thus plays a key role in the global hydrological cycle. In terrestrial ecosystems, plant transpiration accounts for more than 60% of total evapotranspiration and returns approximately 39% of incident precipitation to the atmosphere."*

25. Line 42: exchange "in the movements" with "along the pathway"

**Reply:** We would like to delete this sentence due to changes in the text.

26. Line 43: "but also [due] to ecohydrological separation (Brooks et al., 2010) [and] water isotope …"

**Reply:** We plan to delete all content regarding the TWW hypothesis (see our response to major point 4).

27.  Line 44-45: add reference to Chen et al. (2020) paper

**Reply:** We will add the reference in the next version as suggested.

28.  Line 57: put "respectively" at the end of the sentence?

**Reply:** We will revise this sentence in the next version as suggested, as follows:

*"Poca et al. (2019) reported that arbuscular mycorrhizal fungi can enhance isotopic fractionation during RWU, resulting in up to −24.6‰ and −2.9‰ differences in $\delta^2H$ and $\delta^{18}O$ values, between soil and plant trunk water, respectively."*

29.  Line 60: "incomplete extraction of water during cryogenic distillation could fractionate water isotopes" due to Rayleigh fractionation during the extraction process an incomplete extraction could not only fractionate water stable isotopes in the sample but surely does!

**Reply:** We will revise this sentence in the next version as suggested:

*"For example, incomplete extraction of water during cryogenic distillation fractionates water stable isotopes (Gaj et al., 2017; Orlowski et al., 2018)."*

30.  Line 61-62: "between [cryogenically extracted] stem water and source water"

**Reply:** We will adjust the sentence as suggested:

*"Chen et al. (2020) found the common presence of significant isotopic deviations between cryogenically extracted trunk water and source water in nine woody plant species and demonstrated that this offset stems from methodological artifacts during cryogenic vacuum extraction."*

31. Line 63: "cryogenic extraction-associated methodological artifact" sounds a bit overly complicated

suggestion: "methodological artifacts during cryogenic vacuum extraction"

**Reply:** We will adjust the sentence as suggested:

*"Chen et al. (2020) found the common presence of significant isotopic deviations between cryogenically extracted trunk water and source water in nine woody plant species and demonstrated that this offset stems from methodological artifacts during cryogenic vacuum extraction."*

32. Line 65: "specific process[es]"

**Reply:** We plan to amend this sentence, as follows:

*"Explanation of the isotopic offset between soil and trunk water is essential, but identifying roles of specific processes is generally hindered by the diversity of mechanisms that may be involved."*

33. Line 67: "along [the] soil-root-xylem continuum

**Reply:** We will adjust the sentence as suggested:

*"Moreover, these mechanisms tend to have strongly interactive effects and may act on any compartment along the soil-root-trunk continuum such as soil matrix or soil-root interface or plant woody tissues."*

34. Line 68: "leading to the variation in water isotopes" I don't understand the statement of this subordinate clause.

**Reply:** We will rephrase this sentence and delete the subordinate clause, as follows:

*"Moreover, these mechanisms tend to have strongly interactive effects and may act on any compartment along the soil-root-trunk continuum such as soil matrix or soil-root interface or plant woody tissues."*

35. Line 69: "roots preferentially use tightly bound water according to the TWW hypothesis"

In my opinion this statement is misleading as it might be attributed to the long-term availability of water in the soil. Mobile water (transported in big pores) percolates quickly below the plant rooting zone and therefore is only shortly plant available (during precipitation when rh is usually high and solar radiation low). Hence, roots do not prefer the more tightly bound water, they just use the water they have access to. I know however that it is termed like this in the TWW papers and different people might interpret this statement differently.

**Reply:** We plan to delete all content regarding the TWW hypothesis (see our response to major point 4).

36. Line 78-79: I would delete "during water transport from root to xylem" as you did not measure isotopes during the transport but at two locations (roots and trunk xylem)

**Reply:** We will delete the clause in the next version as suggested.

37. Line 80-82: We hypothesize that 1) there is an isotopic deviation between xylem water of S. matsudana trees and their potential water sources, and that 2) this deviation might be due to a combination of multiple factors.

I think the second hypothesis is quite unspecific and that this is the case is already clear from your literature review. Be more precise on what you investigated and how your research contributes to disentangle combination of multiple contributing factors

**Reply:** We will amend the sentence as suggested:

*"We hypothesize that mobile water is separate from bulk soil water isotopically in the soil matrix and trunk water of* S. matsudana *trees isotopically deviates from their potential water sources due to the heterogeneity of soil water."*

38. Line 98-99: We selected three sampling sites in the check-dammed channel of the Liudaogou catchment. Designated sites 1, 2 and 3 are located 50, 80 and 100 m upstream of the dam, respectively.

**Reply:** We intend to amend these lines as follows:

*"We selected three sampling sites in the check-dammed channel of the Liudaogou catchment, designated sites 1, 2 and 3, located 50, 80 and 100 m upstream of the dam, respectively."*

39. Line 100: "was chose for sampling tree" correct grammar

**Reply:** We intend to amend these lines as follows:

*"Salix matsudana Koidz, is one of the main tree species in the check-dammed catchment, so we chose* S. matsudana *as the sampling tree".*

40. Line 101: "includes" do you mean "consists of"?

**Reply:** Yes, we will amend the sentence as follows:

*"The soil at the site consists of sandy loam and loam according to the USDA classification system, with bulk density ranging from 1.4 to 1.6 g cm$^{-3}$."*

 Line 102: "different soil [depths]" in soil science layers refer to differences in stratification, "in a sampling plot" which sampling plot? As you introduced the numbers before you could just specify

**Reply:** We intend to clarify this, as follows:

*"Water retention curves at 20, 30, 50, 100 and 150 cm soil depths at sampling site 1 are shown in Fig. S1."*

42. Line 104: "from the sampling plot[s]" or "sampling area"? or are you referring to a specific plot?

**Reply:** We intend to clarify this, as follows:

*"Meteorological data on precipitation and air temperature (with 30-min resolution) were obtained from a weather station located about 500 m from sampling site 1."*

43. Line 109: We collected root samples of [one] *S. matsudana* tree at each [of the three] sampling site[s]

**Reply:** We intend to clarify these points as follows:

*"We collected root samples from one* S. matsudana *tree and soil samples at selected soil depths (0-160 cm with 20 cm intervals) at each of the three sampling sites"*

44. Line 112-113: "was collected and measured its isotopic composition" correct grammar

**Reply:** We will correct this sentence, as follows:

*"2-3 coarse roots (> 2 mm diameter) from each sub-cuboid were randomly selected and the roots from the top few centimeters of the topsoil were not artificially removed."*

45. Line 113: "collected disturbed soil samples at 0-160 cm depths" at what interval? Table one suggests 10 cm increments

**Reply:** We will clarify this, as follows:

*"Moreover, we collected disturbed soil samples at 10 cm intervals from 0 to 100 cm depths and 20 cm intervals from 100-160 cm depths using a soil auger to measure soil particle size at sampling site 1."*

46. Line 114-116: Did you measure particle size for both disturbed and undisturbed samples? Or was this only done for disturbed samples and samples in cutting rings were used for water retention curves?

**Reply:** We will clarify this, as follows:

*"We also collected undisturbed soil samples at 20, 30, 50, 100 and 150 cm depths using cutting rings (100 cm³ volume) to obtain water retention curves at the same sampling point."*

47. Line 119: "Our previous results have shown" This reads as it was referring to a previous (published) study. Either you are missing a reference here or you should rephrase the sentence to make clear that this refers to a previous campaign and is up to now unpublished data

**Reply:** We will clarify this, as follows:

*"Previously unpublished data we obtained have shown that the isotopic composition of trunk water of* S. matsudana *trees did not match bulk soil water in the dual-isotope space from May to September 2018."*

48. Line 120-121: I am having difficulty to understand this sentence (referring to TWW)

**Reply:** We will delete this sentence in the next version because it is related to the TWW hypothesis (see our response to major point 4).

49. Line 126: delete "For these analyses" otherwise it reads like precipitation samples are connected to beforementioned sample analysis

**Reply:** We will delete "for these analyses" in the next version as suggested.

*"Precipitation samples were collected as soon as a rain event ended from a polyethylene funnel and bottle, with a plastic ball placed in the funnel to reduce evaporation."*

50. Line 124: "within the period when mobile water was available [ i.e. from August 4 to September 15 2019].

**Reply:** We will amend the sentence accordingly, as follows:

*"So, high frequency sampling (ca. 3-day temporal resolution) was applied to analyze the causes and locations of isotopic deviation during the period when mobile water was available (i.e. from August 4 to September 15 2019)."*

51. Line 128: do you know the depth of the groundwater table at the study site? At which depth did you sample groundwater?

**Reply:** We will add details about the groundwater table in the next version as suggested:

*"At our study site, the mean groundwater table depth was 3.6 m and groundwater samples were collected at ca. 30 cm depth from its surface."*

52. Line 129-131: How were soil samples (for isotopic analysis) stored? How was evaporation from samples prevented?

**Reply:** We intend to clarify these points as follows:

*"These soil samples were also rapidly placed in 10 mL vials that were sealed in the same manner as the root samples, then kept in a cool box until storage in the lab at −20 ℃."*

53. Line 138: "placed in 10 mL vials and wrapped in parafilm" I guess you sealed the vials with caps and then secured the caps with parafilm? Parafilm is not 100% gas tight but permeable to water vapor.

**Reply:** We intend to clarify this, as follows:

*"Pieces of the de-barked and de-leaved twigs, 30 mm long, were then immediately placed in 10 mL vials, the vials were sealed with caps then the caps were secured with Parafilm. These samples were also kept in a cool box until storage in the lab at 4℃."*

54. Line 141-142: "Similarly, of 30 mm long pieces of the de-barked twigs were immediately placed in 10 mL vials and wrapped in parafilm." Is this a repetition or did you want to specify the sampling in different tree heights? Maybe streamline to avoid redundancies

**Reply:** This sentence describes the collection of trunk water at different sampling heights. We will revise these sentences to make them clearer, as suggested.

*"Tree samples were collected simultaneously with the soil samples. These consisted of twigs collected from the south-facing side of three* S. matsudana *trees at 250 cm height on each sampling occasion. In addition, samples of trunk at selected tree heights (150, 250, 350, 450 cm) were collected on August 18, 2019. Bark and phloem were peeled from fully suberized branches to avoid perturbance of trunk water isotopic composition by fractionation. Pieces of the de-barked and de-leaved twigs, 30 mm long, were then immediately placed in 10 mL vials, the vials were sealed with caps then the caps were secured with Parafilm. These samples were also kept in a cool box until storage in the lab at 4℃."*

55. Line 163: b does not appear in your lc-excess formula (is essentially the lc-excess)

**Reply:** We thank the reviewer for alerting us to this error and we will revise the lc-excess formula in the next version, as follows:

"$lc - excess = \delta^2 H_s - a\delta^{18}O_s - b$ "

56. Line 166-167: In my opinion you do not need the link to the TWW here

**Reply:** We will delete this sentence because of the removal of references to TWW hypothesis (see our response to major point 4).

57. Line 174: "To compare the isotopic composition of root and soil water at the same depth"

Actually you only indirectly compare root and bulk soil water with this. If you aim at comparing those two, I think there is better, different approaches. I would rather say you compare their contribution to xylem water. See also my major comment 5) here.

**Reply:** We will delete this sentence because of the removal of references to SIAR model (see our response to major point 6).

58. Line 177: "Based on results of the soil water and root water isotope analysis"

I do not get what the decision criteria was here. Please specify.

**Reply:** We will delete this sentence because of the removal of references to SIAR model (see our response to major point 6).

59. Line 178: "soil [and root] water sources were divided into …"

Then you can delete the next sentence

**Reply:** We will delete this sentence because of the removal of references to SIAR model (see our response to major point 6).

60. Line 181: "because plant water use does not generally cause fractionation of hydrogen and oxygen isotopes"

This contradicts with the statement in your paper that fractionation during rwu influences the xylem isotope values. I would rephrase and write: "assuming no fractionation during plant water uptake"

**Reply:** We will delete this sentence because of the removal of references to SIAR model (see our response to major point 6).

61. Line 202: "overlapped with bulk soil water (Fig 2d)" Actually one needs to look at Figure 2a and d to see the overlap

**Reply:** We intend to clarify this in the next version, as suggested.

62. Line 208: unit ‰ is missing

**Reply:** The information expressed in this sentence is limited, thus we would like to delete it, but we plan to add another table (see Table 1 below), showing the water stable isotopes and lc-excess values for all water samples.

Table 1 Water stable isotopes and lc-excess values for all water samples. Range values show min, max (mean).

| Water samples | N | $\delta^2H$ range (‰) | $\delta^{18}O$ range (‰) | lc-excess range (‰) |
|---|---|---|---|---|
| Groundwater | 22 | −64.7, −63.2 (−64.1) | −9.1, −8.6 (−8.8) | −3.2, −1.0 (−2.4) |
| Mobile water | 191 | −71.7, −48.8 (−61.9) | −10.7, −6.9 (−8.7) | −5.7, 4.6 (−1.2) |
| Bulk soil water | 203 | −89.5, −38.1 (−64.5) | −11.9, −5.1(−8.3) | −12.5, −1.7(−6.7) |
| Less mobile water | 176 | −99.9, −24.6 (−65.1) | −11.2, −2.4 (−8.0) | −23.9, −2.8 (−9.9) |
| Root water | 156 | −71.3, −43.9 (−63.3) | −8.9, −6.5 (−7.6) | −16.9, −2.1 (−10.7) |
| Trunk water | 61 | −70.4, −62.8 (−66.7) | −8.4, −7.3 (−7.7) | −17.1, −9.0 (−13.5) |

63. Line 211: I would not mention TWW here, as it is part of the discussion

**Reply:** We will delete this sentence because of the removal of references to TWW hypothesis (see our response to major point 4).

64. Line 212-213: "suggesting that xylem water was isolated from all potential water sources."

Actually, it only means that xylem water did not reflect bulk soil water sources and not that it is independent from all potential water sources. Also contradicts with the next sentence and is strictly speaking already discussing the results.

**Reply:** We would like to delete this sentence, but add a brief discussion about this issue in Section 4.1.2 of the Discussion "Isotopic offset between bulk soil water and root water", as follows:

*"The most plausible explanation for isotopic mismatch between root water and bulk soil water in dual-isotope plots is that bulk soil water is not representative of available plant water sources because of the heterogeneity of bulk soil water. As shown in Fig. 1, less mobile water overlapped isotopically with root water after removing the influence of mobile water. The rapidity of mobile water's passage through soil reduces its contact with mineral surfaces, and hence its nutrient concentrations (McDonnell, 2017; Sprenger et al., 2019). Thus, plants may have used large amounts of less mobile water that was strongly affected by evaporative effects in the presented study, isotopically distinct from mobile water and groundwater, and with similar isotopic composition to trunk water."*

McDonnell, J.J.: Beyond the water balance, Nat. Geosci., 10, 396-396, 2017.

Sprenger, M., Llorens, P., Cayuela, C., Gallart, F., and Latron, J.: Mechanisms of consistently disjunct soil water pools over (pore) space and time. Hydrol. Earth Syst. Sci., 23, 2751-2762, 2019.

65. Line 213-214: streamline sentence to avoid repetition of tightly bound water, also includes discussion of results already

**Reply:** We will delete this sentence because it is more suitable for the *Discussion* than the *Results*.

66. Line 232: "[horizontal] distance"

**Reply:** We will amend the sentence as suggested:

*"There were no significant differences (p > 0.05) in isotopic composition ($\delta^2H$ and $\delta^{18}O$) of either root water or bulk soil water between 40 cm and 80 cm horizontal distance."*

67. Line 233: exchange "> 80 cm" with "within 80 cm"

**Reply:** We will amend the sentence as suggested:

*"There were no significant differences (p > 0.05) in isotopic composition ($\delta^2H$ and $\delta^{18}O$) of either root water or bulk soil water between 40 cm and 80 cm horizontal distance, suggesting that isotopic composition of the soil was horizontally homogenous within 80 cm from tap roots."*

68. Line 239 & 241: $\delta^2O$ should be $\delta^{18}O$ I assume, also stay consistent with the subscripts, here the subscript soil refers to bulk soil water but subscript BW also exists

**Reply:** We thank the reviewer for alerting us to this error and will revise this sentence in the next version as follows:

*"Similarly, a strong correlation was observed between $\Delta^{18}O$ ($\Delta^{18}O = \delta^{18}O_{soil} - \delta^{18}O_{trunk}$) and $\Delta^2H$ ($\Delta^2H = \delta^2H_{soil} - \delta^2H_{trunk}$) soil-trunk offsets during August 4 to September 15."*

69. Line 243: "These results show that water isotopes, especially hydrogen isotopes, changed between…"

Water stable isotopes do not change, the ratios of the isotopes change

**Reply:** We will delete this sentence because of the changes in content.

70.   Line 244: "supporting our first hypothesis" belongs into discussion

**Reply:** We will delete this sentence in the next version because it is related to the TWW

hypothesis (see our response to major point 4).

71.   Line 246: Specify heading of subsection, contribution of what?

**Reply:** We would like to delete Section 3.4 of the Results, "Contributions", because of the

removal of references to the SIAR model (see our response to major point 6).

72.   Line 248: "Potential sources of plant xylem water were determined using…"

you do not determine potential sources but the contribution of these sources to rwu

**Reply:** We will delete this sentence because of the removal of references to SIAR model (see our

response to major point 6).

73.   Line 256: "Separation of mobile water and bulk soil water in the soil matrix"

Your manuscript deals with this separation a lot and it is also a big part of your study design. Why

is this not one of your hypothesis/aims?

**Reply:** Thanks for your suggestion. We will change it to one of our hypotheses in the next version,

as follows:

*"We hypothesize that mobile water is isotopically separate from bulk soil water in the soil matrix*

*and isotopic deviation occurs between trunk water of* S. matsudana *trees and their potential water*

*sources due to heterogeneity of the soil water."*

74.   Line 257: exchange "covered" with "experimental"

**Reply:** We will revise this sentence in the next version as suggested:

*"At our study site during the experimental period (August 4 to September 15, 2019), a clear*

*isotopic separation between mobile and bulk soil water was observed."*

75. Line 261: "clear [isotopic] separation"

**Reply:** We will add "isotopic", as suggested:

*"The lc-excess values of mobile and bulk soil water consistently differed significantly, although*

*GWC varied greatly, suggesting a clear isotopic separation between mobile and bulk soil water*

*that is not affected by GWC."*

76. Line 266-268: Reference for statement missing

**Reply:** We will add references in the next version, as follows:

*"Gierke et al. (2016) examined the stable isotopic composition of precipitation, bulk soil water*

*and trunk water in a high elevation watershed and their results suggested that mobile water was*

*primarily associated with summer thunderstorms, and thus subject to minimal evaporative loss. In*

*contrast, less mobile water was derived from snowmelt, filling small pores in the shallow soils.*

*Allen et al. (2019) characterized the occurrence of winter and summer precipitation in plant trunk*

*samples using a seasonal origin index and found that winter precipitation was the predominant*

*water source for midsummer transpiration in sampled beech and oak trees. Due to seasonal*

*isotopic cycles in precipitation, there may be clear distinctions in the isotopic composition of*

*mobile water and less mobile water derived from precipitation falling at different times (Bowen et*

*al., 2019)"*

Allen, S.T., Kirchner, J.W., Braun, S., Siegwolf, R.T.W., and Goldsmith, G.R.: Seasonal origins of

soil water used by trees, Hydrol. Earth Syst. Sci., 23, 1199-1210, 2019.

Bowen, G.J., Cai, Z.Y., Fiorella, R.P., and Putman, A.L.: Isotopes in the water cycle: regional- to

global-scale patterns and applications, Annu. Rev. Earth Planet. Sci., 47, 453-479, 2019.

Gierke, C., Newton, B.T., and Phillips, F.M.: Soil-water dynamics and tree water uptake in the Sacramento Mountains of New Mexico (USA): a stable isotope study, Hydrogeol. J., 24, 805-818, 2016.

77. Line 268-269: "Due to the seasonal variation in precipitation, winter and summer precipitation have different isotope signals"

Both half sentences state the same thing. Seasonal variation is not the cause for isotopic differences between winter and summer precipitation

**Reply:** Following the suggestion we will amend this sentence, as follows:

*"Due to seasonal isotopic cycles in precipitation, there may be clear distinctions in the isotopic composition of mobile water and less mobile water derived from precipitation falling at different times (Bowen et al., 2019)."*

78. Line 272: place "at our study site" at the beginning of the sentence

**Reply:** We will amend the sentence as suggested:

*"At our study site, precipitation in winter (December-February) and summer (June-September) accounted for 2% and 77% of total average annual precipitation (464 mm) from 2003 to 2019, respectively."*

79. Line 274: "caused by other factors, and not necessarily by seasonal variation in precipitation"

Observed differences could also stem from isotopic differences of individual precipitation events.

Did you check if with high and low intensities are systematically different? If so, high intensity precipitation events might percolate faster into the soil and also contribute bigger ´water quantities to sampled mobile water.

**Reply:** Thanks for your suggestions. In the next version we would like to change our discussion of this issue based on the antecedent precipitation events, as follows:

*"Notably, there was a major rainstorm the day before the sampling (August 3), with 63 mm precipitation. The mean GWC in 0-50 and 100-150 cm layers reached 17.4 ± 2.7% and 10.8 ± 1.5% between August 4 and August 7, respectively. These results imply that precipitation greatly supplemented water in the upper soil layer. So mobile water collected by suction lysimeters during this period contained a considerable proportion of water from the rain event on August 3. In contrast, bulk soil water contained not only mobile water from this rain event, but also antecedent less mobile water that could not be extracted by a suction lysimeter, resulting in the isotopic separation between mobile water and bulk soil water."*

80. Line 277: "with [increasing] soil depth"

**Reply:** We will add "increasing" as suggested:

*"The effect of soil evaporation on bulk soil water gradually weakens with increasing soil depth."*

81. Line 281-282: "Although the mixing of mobile and tightly bound water conflicts with the original hypothesis of Brooks et al. (2010)"

Does it really disagree? Or is it more a question of the degree of mixing or rather that mixing between soil water pools is not complete as previously assumed?

**Reply:** We will delete this sentence because of the removal of references to TWW hypothesis (see our response to major point 4).

82. Line 282-285: Increased mixing with soil depth was e.g. also observed/mentioned by Sprenger et al. (2016) and Kübert et al. (2020)

https://agupubs.onlinelibrary.wiley.com/doi/full/10.1002/2015RG000515

https://www.frontiersin.org/articles/10.3389/fpls.2020.00387/full

**Reply:** Thanks for this suggestion. We will add these references in the next version, as follows:

*"Both mobile water and less mobile water in deep layers are more fully mixed than in shallow layers (Sprenger et al., 2016; Kubert et al., 2020)."*

83. Line 290-291: "driven by spatiotemporal dynamics of soil water profiles associated with soil evaporation"

What do you mean with that? The temporal (and spatial) differences in infiltration and evaporation and how those influence the sampled soil water sources?

**Reply:** We will delete this sentence because of the removal of references to TWW hypothesis (see our response to major point 4).

84. Line 296-297: "These results showed that the isotopic offset between plant root water and soil water occurred at the root-soil interface."

I think this statement is misleading. I think this statement is misleading. In my opinion it points towards the explanation that bulk soil water is not representative of available plant water sources (see next paragraph). This heterogeneity does however not only apply to the root-soil interface. Or am I missing something here?

**Reply:** As mentioned in our response to comment 2, we would like to revise this sentence in the next version as follows:

*"These findings are not consistent with the greater $^2H$ enrichment in root water than in bulk soil water (differences up to 8.6‰) we detected, suggesting that soil-root isotopic offsets are more likely to be caused by the complexity of root systems and heterogeneity of bulk soil water than isotopic fractionation during root water uptake."*

85. Line 302: "that is strongly affected by evaporative effects [in the presented study], isotopically separated from mobile water and groundwater and shows similar enriched isotopic signals [than xylem water]"

**Reply:** We will amend the sentence as suggested:

*"Plants may use large amounts of less mobile water that is strongly affected by evaporative effects in the presented study, isotopically separated from mobile water and groundwater and shows similar enriched isotopic composition than trunk water."*

86. Line 303-304: "This hypothesis is corroborated by the overlap in isotopic composition between root and bulk soil water at 0-60 cm depths (Fig. 2 and 5)."

Why? As I understand it, this speaks against the statement in the sentence before.

**Reply:** We will delete this sentence in the next version because it is related to the TWW hypothesis (see our response to major point 4).

87. Line 304-309: "We considered whether bulk soil water isotopes can represent isotopic values of tightly bound water used by plants. Generally, the water designated 'bulk soil water' includes mobile and tightly bound water due to limitations of water extraction technology when assessing the TWW hypothesis. Thus, the proportion of mobile water in the bulk soil water increases as soil moisture increases, resulting in isotopic deviation between root water and bulk soil water."

I do not understand the argument here. Maybe rephrasing helps :-D

**Reply:** We will delete this sentence in the next version because it is related to the TWW hypothesis (see our response to major point 4).

88. Line 331-332: "Under the assumption that plant fractionation does not occur"

Do you mean that the isotopic composition of water does not change during within plant

transport? Fractionation is mainly believed to occur during rwu I think. However, as you sampled

the roots, you eliminate this influencing factor.

**Reply:** We will delete this sentence in the next version because of the changes in content.

89.  Line 341: "also has residence times in branches and roots"

What do you mean with this? How is it different from the time lag due to transport from roots to

branches? Do you refer to an influence of xylem water storage?

**Reply:** We intend to clarify this, as follows:

"*As the time required for isotopic tracer ($D_2O$) to move from the base of a trunk to the upper*

*crown of a tree reportedly ranges from 2.5 to 21 days (Meinzer et al., 2016), the isotopic*

*composition of trunk water may differ from that root water collected on the same day (August*

*18).*"

Meinzer, F.C., Woodruff, D.R., Marias, D.E., Smith, D.D., Mcculloh, K.A., Howard, A.R., and

Magedman, A.L.: Mapping 'hydroscapes' along the iso- to anisohydric continuum of stomatal

regulation of plant water status, Ecol. Lett., 19, 1343-1352, 2016.

90.  Line 346: "that isotope enrichment may have been present in the unsampled branches"

Why would you measure a potential enrichment caused by unsampled branches by sampling at

different heights? What do you mean with enrichment present in unsampled branches?

**Reply:** We will revise this sentence in the next version as follows:

*"Moreover, to test the possibility that isotopic composition of trunk water may be heterogeneous at different tree heights, we collected trunk water at 150-450 cm tree heights on August 18, 2019, and found no significant differences ($p > 0.05$) (Fig. S3)."*

91.  Line 348: "xylem water was [isotopically] more enriched than…"

**Reply:** We will amend the sentence as follows:

*"Furthermore, previous studies have provided indications that trunk water becomes more enriched in $^{18}O$ due to the temporal declines in sap flow rates (Martin-Gomez et al., 2017) and the mixture of trunk water with leaf water (Brandes et al., 2007)"*

Brandes, E., Wenninger, J., Koeniger, P., Schindler, D., Rennenberg, H., Leibundgut, C., Mayer, H., and Gessler, A.: Assessing environmental and physiological controls over water relations in a Scots pine (*Pinus sylvestris* L.) stand through analyses of stable isotope composition of water and organic matter. Plant Cell Environ., 30,113-127, 2007.

Martin-Gomez, P., Serrano, L., and Ferrio, J.P.: Short-term dynamics of evaporative enrichment of xylem water in woody stems: implications for ecohydrology, Tree Physiol., 37, 511-522, 2017.

92.  Line 352-353: "However, we found that the xylem water contained [more of the depleted] isotopic signal of deep roots than [of the] enriched signal from shallow roots. The results show that there was no isotopic fractionation during water transport from root to xylem"

I am not sure if this allows the conclusion that no fractionation during transport took place.

However, I agree that it strongly suggests it. I would also specify here that you did not observe an enrichment (fractionation could go both ways) during transport as other authors suggested.

**Reply:** We plan to amend the sentence as follows:

*"Furthermore, previous studies have provided indications that trunk water becomes more enriched in $^{18}O$ due to the temporal declines in sap flow rates (Martin-Gomez et al., 2017) and the mixture of trunk water with leaf water (Brandes et al., 2007). However, we did not find that trunk water of the trees we sampled had higher $\delta^{18}O$ values than root water, Thus, we believe it reflects the selective utilization of water source rather than isotopic fractionation within woody tissues."*

Brandes, E., Wenninger, J., Koeniger, P., Schindler, D., Rennenberg, H., Leibundgut, C., Mayer, H., and Gessler, A.: Assessing environmental and physiological controls over water relations in a Scots pine (*Pinus sylvestris* L.) stand through analyses of stable isotope composition of water and organic matter. Plant Cell Environ., 30,113-127, 2007.

Martin-Gomez, P., Serrano, L., and Ferrio, J.P.: Short-term dynamics of evaporative enrichment of xylem water in woody stems: implications for ecohydrology, Tree Physiol., 37, 511-522, 2017.

93. Line 358: exchange "covered" with "experimental" or write "period covered"

**Reply:** We will revise this sentence in the next version as suggested:

*"At our study site during the experimental period, the isotopic offset existed between trunk water of* S. matsudana *trees and bulk soil water."*

94. Line 359-360: "isotopic offset [exists] between xylem water and [bulk soil water]"

As you elaborate in your manuscript, bulk soil water might not reflect all available soil water sources.

**Reply:** We would like to rephrase the Conclusion in the next version because of the changes of in conclusions such as those regarding the TWW hypothesis (see our response to the major point 4) and plant fractionation (see our response to major point 2), as follows:

*"At our study site during the experimental period, there was an isotopic offset between trunk water of* S. matsudana *trees and bulk soil water. We explored causes of the mismatch and sources of water taken up by the trees by analyzing the stable isotope composition of soil water with various mobility, root water and trunk water. In the soil matrix, bulk soil water generally had lower lc-excess values than mobile water, due to effects of soil evaporation and mixture of newly infiltrated mobile and less mobile water with increasing depth. Root water did not match bulk soil water at the same depth completely, due to the complexity of root systems and soil water heterogeneity. The maximum differences in $\delta^2H$ and $\delta^{18}O$ between bulk soil water and root water were $-8.6$ and $-1.8‰$, respectively. Overall, the $\delta^2H$ and $\delta^{18}O$ values derived for less mobile water overlapped with those of root water and trunk water, and the trunk water values mainly overlapped with those of root water at 100-160 cm depths. These findings suggest that the isotopic offset between bulk soil water and trunk water was due to isotopic mismatch between root water and bulk soil water associated with heterogeneity of the soil water. The presented stable isotope data for bulk soil water, mobile water, less mobile water, root water and trunk water were highly valuable for analyzing the spatial heterogeneity of water fluxes in the root zone, and elucidating the water sources used by the plants."*

95. Line 361: "and water flow paths" what do you mean with that? Infiltration along preferential flow paths?

**Reply:** We intend to clarify this, as follows:

*"In the soil matrix, bulk soil water generally had lower lc-excess values than mobile water, due to effects of soil evaporation and mixture of newly infiltrated mobile and less mobile water with increasing depth."*

96.   Line 369: "the [estimated] contribution of roots in these depths to xylem water [was] 74%."

**Reply:** We will delete this sentence because of the removal of references to SIAR model (see our response to major point 6).

---

## Author Comment (AC2)

Response to Reviewer 2, re: "Insights into isotopic mismatch between soil water and *Salix matsudana* Koidz xylem water from root water isotope measurements", in review in HESSD (NO. hess-2020-680).

We thank Reviewer #2 for the time and effort spent providing us with valuable feedback, which can certainly help us to strengthen the manuscript. We agree with the suggestions and have made targeted amendments, as detailed in the point-by-point responses below. The reviewer's comments are presented in blue, while passages changed in specific responses to the comments are presented in quotation marks and italic font.

**Point-by-point by responses to Reviewer 2's comments**

**Major points:**

1. In the manuscript entitled, "Insights into isotopic mismatch between soil water and *Salix matsudana* Koidz xylem water from root water isotope measurements", results from a field study show isotopic differences between different soil water and plant water pools. The dataset is interesting and results from intensive measurements. I think some of the findings are potentially interesting and warrant publication. However, the current presentation has some weaknesses and ambiguities that need to be addressed. Ultimately, the imprecise use of jargon obscures the interpretation and implications of the study.

**Reply:** We thank the reviewer for the constructive comments and suggestions, we have carefully considered them and tried our best to reduce the highlighted weaknesses and ambiguities in the

manuscript, especially the problem about the "two water worlds" hypothesis.

2. Throughout, references are made to ecohydrological separation and the two-water-worlds

(TWW) hypothesis. However, I am not sure how the authors define these phenomena; their definitions seem different from my own, different from the literature that they are citing, and they may change throughout. At times, it seems that these terms just mean "there are isotopic differences", which is not particularly novel to identify. The paper would greatly benefit from less use of jargon. It should be more explicitly stated what is being tested. From my reading of this, the key questions of this paper are "Does root water isotopic composition match that of soils' bulk, mobile, and bound water fractions of soils at the same depths". Then there is a second question "How does choice of stem sample (from four heights) or choice of soil versus roots influence inferences of water uptake depths". Neither of these questions is especially related to TWW or ecohydrological separation. Frankly, the duration of the study is too short to assess either TWW or ecohydrological separation because any observed differences between plant water and groundwater isotope ratios might be a product of lags, rather than the use of fundamentally different sources. The process Brooks et al referred to as ecohydrological separation was only observable through using measurements across multiple seasons.

**Reply:** We agree that the short experimental period and the focus on phenomena that are not directly related to the "two water worlds" (TWW) hypothesis hinder concise, meaningful discussion of the hypothesis and ecohydrological separation. Reviewer 1 also suggested that references to the TWW hypothesis should be reduced. Therefore, we would like to delete all content about the TWW hypothesis and pay more attention to the soil water's heterogeneity

through the comparison of mobile water, bulk soil water, and derived characteristics of less mobile water (we intend to change 'tightly bound water' to 'less mobile water', as advised by Reviewer 1) at the same depths, and the impact of this heterogeneity on plant water uptake in the next version.

3. The isotopic differences between root water and soil water is key to the conclusions made in this paper. The authors seem to suggest that the water in roots should match the water in soils around them. They did not, and this was interpreted as potential fractionation. However, roots can transport water from different locations. I would only expect similarity between roots and surrounding soils if fine roots were sampled. For coarser roots, as used in this study, I would expect those roots to transport water from much deeper depths and integrate large volumes of soil water. Thus, it is not clear that "a combination of plant fractionation and TWW-type separation" (which, again, needs to be clarified) is needed to explain the observations here. This needs to be further discussed. Potentially, additional excavations may be warranted to identify whether the size of root samples could include fine roots that extend substantially deeper.

**Reply:** Thanks for your suggestions. We would like to change the conclusion that isotopic fractionation leads to the observed mismatch between root water and bulk soil water at the same depth in the manuscript, for two reasons. First, the water in the sampled coarse roots (> 2 mm diameter) does not necessarily match the bulk soil water around them because sampled coarse roots can transport and mix water from different locations, as suggested. Second, as suggested by Reviewer 1, recent studies on isotopic fractionation have found stronger 2H depletion in trunk water/root water than in bulk soil water (e.g., Poca et al., 2019; Vargas et al., 2017). However,

these findings are not consistent with our finding that root water had higher  $\delta^2$ H values than bulk soil water (up to 8.6‰). Most importantly, we found that the isotopic composition of root water deviated from that of bulk soil water, but overlapped with the values derived for less mobile water (see Figure 1 below). Thus, we concluded that soil-root isotopic offsets are more likely to be caused by the complexity of root systems and the heterogeneity of bulk soil water than isotopic fractionation during root water uptake. Hence, we would like to add the following discussion regarding this issue in the next version:

[revised manuscript text omitted]

---

## Author Response (AR1)

**Reply to the comments from the Reviewers and Editor on the manuscript (NO. hess-2020-680) "Insights into isotopic mismatch between soil water and *Salix matsudana* Koidz xylem water from root water isotope measurements".**

We thank the Editor Matthias Sprenger and both Reviewers for their thoughtful and critical reviews of our manuscript. We greatly appreciate the positive feedback and many well-founded points that have certainly helped us to improve the manuscript. We have revised it according to the expressed concerns and comments, as noted in our point-by-point responses.

To facilitate assessment of our revision, we have listed below first the main eight changes in the revised manuscript, then provided detailed point-by-point replies to the two reviewers' comments. The reviewer's comments are presented in blue. Passages changed in specific responses to the comments are presented (together with page and line number) in quotation marks and italic font. All the changes in the revised manuscript have been marked by change-tracking.

**The main changes are:**

**(1) All contents about the "two water worlds" (TWW) hypothesis have been deleted in the revised manuscript following advice of the Reviewers #1 and #2.**

Reviewer #1 suggested that references to the TWW hypothesis should be reduced. Moreover, the short experimental period and the focus on phenomena that are not directly related to the TWW hypothesis hinder concise, meaningful discussion of the hypothesis and ecohydrological separation, as pointed out by Reviewer #2. Therefore, we have deleted all contents about the TWW hypothesis

and paid more attention to the soil water's heterogeneity through the comparison of mobile water, bulk soil water, and derived characteristics of less mobile water at the same depths, and the impact of this heterogeneity on plant water uptake in the revised manuscript.

**(2) The reason for the observed mismatch between root water and bulk soil water has been changed from isotopic fractionation to bulk soil water heterogeneity and the complexity of root systems, as advised by the Reviewers #1 and #2.**

We have changed the conclusion that isotopic fractionation leads to the observed mismatch between root water and bulk soil water at the same depth in the revised manuscript, for following reasons. First, recent studies on isotopic fractionation have found stronger $^2$H depletion in trunk water/root water than in bulk soil water (e.g., Poca et al., 2019; Vargas et al., 2017). However, these findings are not consistent with our finding that root water had higher $\delta^2$H values than bulk soil water (up to 8.6‰) as pointed out by Reviewer #1. Second, the water in the sampled coarse roots (> 2 mm diameter) does not necessarily match the bulk soil water around them because sampled coarse roots can transport and mix water from different locations (Reviewer #2). Most importantly, we found that the isotopic composition of root water deviated from that of bulk soil water, but overlapped with the values derived for less mobile water (Figure 3). Thus, we concluded that soil-root isotopic offsets are more likely to be caused by the complexity of root systems and the heterogeneity of bulk soil water than isotopic fractionation during root water uptake.

**(3) Calculation of plant water source contributions based on SIAR model has been deleted, as advised by Reviewer #1.**

**(4) The abstract and conclusion have been rephrased.**

**Abstract:**

Increasing numbers of field studies have detected isotopic mismatches between plant trunk water and its potential sources. However, the cause of these isotopic offsets is not clear and it is uncertain whether they occur during root water uptake or during water transmission from root to trunk. Thus, we measured the specific isotopic composition ($\delta^2$H and $\delta^{18}$O) of each component (e.g., bulk soil water, mobile water, groundwater, trunk water and root water of *Salix matsudana* Koidz trees) with about three-day resolution in the soil-root-trunk continuum. We report three main findings. First, we detected clear separation between mobile water and bulk soil water isotopic composition, but the distinction between mobile water and bulk soil water gradually decreased with increasing soil depth. Second, root water deviated from bulk soil water isotopic composition, but it overlapped with the composition derived for less mobile water. The maximum differences in $\delta^2$H and $\delta^{18}$O between bulk soil water and root water were −8.6 and −1.8‰, respectively. Third, trunk water was only isotopically similar to root water at 100-160 cm depths, and it remained stable during the experimental period, suggesting that the trees consistently used the stable deep water source. In conclusion, the isotopic offset between bulk soil water and trunk water of *S. matsudana* reflected an isotopic mismatch between root water and bulk soil water associated with heterogeneity of the soil water. Our results illuminate relationships between the isotopic composition of soil water of various mobility, root water and trunk water that may be useful for advancing our understanding and representation of root water uptake and transport.

**Conclusion:**

At our study site during the experimental period, there was an isotopic offset between trunk water of *S. matsudana* trees and bulk soil water. We explored causes of the mismatch and sources of

water taken up by the trees by analyzing the stable isotope composition of soil water with various mobility, root water and trunk water. In the soil matrix, bulk soil water generally had lower lc-excess values than mobile water, due to effects of soil evaporation and mixture of newly infiltrated mobile and less mobile water with increasing depth. Root water did not match bulk soil water at the same depth completely, due to the complexity of root systems and soil water heterogeneity. The maximum differences in $\delta^2H$ and $\delta^{18}O$ between bulk soil water and root water were −8.6 and −1.8‰, respectively. Overall, the $\delta^2H$ and $\delta^{18}O$ values derived for less mobile water overlapped with those of root water and trunk water, and the trunk water values mainly overlapped with those of root water at 100-160 cm depths. These findings suggest that the isotopic offset between bulk soil water and trunk water was due to isotopic mismatch between root water and bulk soil water associated with heterogeneity of the soil water. The presented stable isotope data for bulk soil water, mobile water, less mobile water, root water and trunk water were highly valuable for analyzing the spatial heterogeneity of water fluxes in the root zone, and elucidating the water sources used by the plants.

(5) **Details about the root water sampling and schematic diagram of root excavations (Figure 2) in the *Materials and methods* section have been added as advised by Reviewer #1.**

(6) **＂Xylem water" has been changed to "trunk water", as advised by Reviewer #1.**

(7) **＂Bound water" has been changed to "less mobile water" as advised by Reviewer 1. Meanwhile, the less mobile water stable isotope data in the supplemental figures S3-4 have been incorporated into the main figures 3 and 4.**

(8) **Another table (Table 2) has been added, showing the water stable isotopes and lc-excess values for all water samples.**

**Point-by-point by responses to Reviewer 1's comments**

*Major points:*

1. The manuscript hess-2020-680: "Insights into isotopic mismatch between soil water and *Salix matsudana* Koidz xylem water from root water isotope measurements" by Zhao and Wang investigates potential reasons for an observed mismatch between soil and xylem water stable isotope values by measuring water stable isotope ratios in different soil water pools (mobile, bulk soil and bound) across soil depths, as well as in roots (across depth) and xylem of three rigorously sampled tree individuals. The authors conclude that the observed isotopic differences between xylem and bulk soil water arise from a combination of ecohydrological separation, i.e. isotopic differences in mobile and bound soil water and plant fractionation during root water uptake.

   Root water uptake depth is routinely determined by comparing the isotope composition of xylem water with that of soil water in different depths (and other water sources like stream and groundwater) assuming that extracted water from bulk soil samples represents available water sources. This long-standing principle is lately repeatedly questioned and a lot of uncertainty persists on potential reasons and underlying causes. This work contributes to the discussion and provides new insights. I especially liked, that the authors conducted an experiment under natural conditions. I do not know of any other study that sampled water sources, xylem water and also systematically investigated root xylem within a field experiment to tackle this question and I applaud the authors for conducting this surely very labour-intensive work. The manuscript is well-structured and understandable. Generally, I am in favour of publishing this work. However, I think that a number of critical points (see major

points below) should be addressed beforehand in a revised version. I also suggest an English

native speaker to proof-read the manuscript and help to further improve some of the

expressions.

**Reply:** We thank the reviewer for the constructive comments and suggestions, we have carefully

considered them and tried our best to address the highlighted weaknesses in the revised

manuscript. In addition, we have invited a professional native English-speaking editor with a PhD

in a relevant discipline to edit the revised manuscript.

2.  You state that plant fractionation, i.e. change of isotope values during root water uptake, is

    one of two main causes for the observed mismatch. This was observed before in xerophytic

    and halophytic plants and for plants in symbiosis with arbuscular mycorrhiza (e.g. Ellsworth

    & Williams 2007, Poca et al. 2019). However, previous studies on plant fractionation

    reported depleted (more negative) isotope values in plant xylem as compared to soil water,

    hence plants discriminated against the heavy isotope (mostly $^2$H). This is not in line with

    enriched (less negative) xylem values reported here. While it cannot be ruled out completely

    that water would get enriched in heavy isotopes during rwu, this was not reported before and

    considering other potential reasons, namely isotopic heterogeneities across soil water pools as

    well as temporal variability and methodological artefacts, this seems unlikely to me. If you

    decide to keep it in the manuscript, this discrepancy to other studies should be pointed out

    and discussed in detail.

**Reply:** Thanks for your suggestions. We have changed the conclusion that isotopic fractionation

leads to the observed mismatch between root water and bulk soil water at the same depth in the

revised manuscript. Please see the main changes (2). We have added the following discussion

regarding this issue in the revised manuscript (Pages 12-13 lines 375-413):

[revised manuscript text omitted]

2017.

3.    Building on that, some root and all stem xylem samples show an evaporative enrichment in

the dual isotope plot (Fig. 2). This is discussed in the manuscript and related to an enriched

signal of bound soil water. I think this argument would be strengthened if you provide further

description on the sampling procedure. Specifically, I wondered if evaporation during root

sample collection could potentially influence obtained results. Did you sample roots right

from the soil profile wall or exclude the first few centimetres? How fast was sampling

conducted after digging the hole? Also: Was soil thoroughly removed from sampled roots?

Regarding xylem sampling, you write "Bark was peeled from the twigs and all leaves were

removed to avoid perturbance of xylem water isotopic signatures by fractionation." (L 136-

137). So, did you sample twigs that had leaves directly attached to them? Could the enriched

signal hence arise from back-diffusion or any other exchange with enriched leave water

before sampling? Were twigs fully suberized or were they green and hence photosynthetically

active?

**Reply:** We have added details about the root and trunk water sampling and schematic diagram of

root excavations (see Figure 2 below) in the *Materials and methods* section, as follows:

Regarding root sampling (Page 5 lines 134-144):

*"We excavated a soil cuboid with 160 cm depth, 80 cm width (horizontal distance) and 160 cm*

*length with the main root of the selected tree at the center (Fig. 1d and Fig. 2a). We then divided*

*the cuboid into 64 sub-cuboids (length, 40 cm; width, 40 cm; height, 20 cm) (Fig. 2b) and dug*

*each sub-cuboid one by one to minimize risks of evaporation. 2-3 coarse roots (> 2 mm diameter)*

*from each sub-cuboid were randomly selected and roots from the top few centimeters of the*

*topsoil were not artificially removed. To minimize the influence of attached soil on root water,*

*these sampled roots were rapidly peeled to remove bark, placed in 10 mL vials and sealed with*

*caps then the caps were secured with Parafilm. Finally, these samples were kept in a cool box*

*until storage in the lab at 4℃. To compare the isotopic composition of root and bulk soil water at*

*the same depths, we collected samples of soil around the sampled roots in each sub-cuboid. These*

*soil samples were also rapidly placed in 10 mL vials that were sealed in the same manner as the*

*root samples, then kept in a cool box until storage in the lab at −20℃."*

[Figure]

Figure 2: Schematic diagram of root excavations (a) and measurements (b).

Regarding trunk sampling (Page 6 lines 173-179)

*"Tree samples were collected simultaneously with the soil samples. These consisted of twigs collected from the south-facing side of three S. matsudana trees at 250 cm height on each sampling occasion. In addition, samples of trunk at selected tree heights (150, 250, 350, 450 cm) were collected on August 18, 2019. Bark and phloem were peeled from fully suberized branches to avoid perturbance of trunk water isotopic composition by fractionation. Pieces of the de-barked and de-leaved twigs, 30 mm long, were then immediately placed in 10 mL vials, the vials were sealed with caps then the caps were secured with Parafilm. These samples were also kept in a cool box until storage in the lab at 4°C."*

4. In my personal opinion, I would not put so much emphasise on the TWW hypothesis. If a reader is not familiar with it, it might be a bit confusing (especially in the abstract). I think it is good that you refer your results to it but I would reduce the importance it has in your manuscript, e.g the amounts of mentions.

Also, I think the term "tightly bound water" is misleading. At least in my perception it suggests that plants use an exceptionally tightly bound water pool (as compared simply to bound water). However, in my opinion, they probably just use the water that is available to them and mobile water might infiltrate beyond the root zone too quickly to be available in the long term. I imagine that the sampled mobile water (in lysimeters) mainly originates from percolation during precipitation events, when relative humidity is high, solar radiation is low and plants hence do not transpire a lot. I suggest to use "less mobile water" or simply "bound water". However, I am aware that it is also termed "tightly bound water" in the TWW hypothesis.

**Reply:** We agree that references to the TWW hypothesis should be reduced. Moreover, the short experimental period and the focus on phenomenon that are not directly related to the TWW hypothesis hinder concise, meaningful discussion of the hypothesis and ecohydrological separation, as pointed out by Reviewer #2. Therefore, we have deleted all contents about the TWW hypothesis following advice of both Reviewers. In addition, we have changed "bound water" to "less mobile water" in the revised manuscript as suggested.

5. You took the time to calculate the isotopic composition of tightly bound soil water (from measured bulk and mobile soil water) and you repeatedly argue that plants preferentially use this water source. However, this data is only incorporated into one supplemental figure. I think it would strengthen the story if those values were incorporated into the main figures as well (e.g. Figure 2 and 5).

**Reply:** We have added data on the less mobile water stable isotopes in the supplemental figures to the main figures as suggested. Please see Figures 3-4 below.

[Figure]

Figure 3 (a) $\delta^{18}O$ and $\delta^2H$ isotopic composition collected from August 4 to September 15, 2019.

Plotted values include bulk soil water (BW), mobile water (MW), root water (RW), trunk water

(TW), less mobile water (LMW) and groundwater (GW). (b) $\delta^{18}O$ and $\delta^2H$ isotopic composition

of GW, and MW collected from different depths, (c) BW collected from different depths, (d)

LMW collected from different depths, (e) RW collected from different depths, and (f) TW

collected from different tree heights. The red line represents the 2016-2019 local meteoric water

line (LMWL, $\delta^2H = 5.91 + 7.67\,\delta^{18}O$, $R^2 = 0.96$). The black line represents the global meteoric

water line (GMWL, $\delta^2H = 10 + 8\,\delta^{18}O$).

[Figure]

Figure 4 (a-f) Temporal dynamics of hydrological conditions (precipitation and gravimetric water

content, GWC) and lc-excess values (these values are means and standard deviations for three

sites) of groundwater (GW), trunk water (TW), mobile water (MW), less mobile water (LMW) and bulk soil water (BW) at indicated depths (20, 30, 50, 100 and 150 cm) during the period August 3 to September 15, 2019. (A) Boxplots of total MW (N=191), GW (N=22), BW (N=204), TW (N=61) and LMW (N=176) lc-excess values. (B-F) Boxplots of MW and BW at 20 cm (MW, N=40; BW, N=42; LMW, N=39), 30 cm (MW, N=40; BW, N=40; LMW, N=34), 50 cm (MW, N=38; BW, N=40; LMW, N=33), 100 cm (MW, N=36; BW, N=40; LMW, N=34) and 150 cm (MW, N=37; BW, N=42; LMW, N=36) depths. The top and bottom of each box are the 25th and 75th percentiles of the samples, respectively. The black line in each box is the sample median. Trunk water and potential water sources that do not share a letter are significantly different (p < 0.05, Tukey-Kramer HSD).

6.   SIAR modelling: In my opinion, this does not strengthen the story. The isotopic composition of water sources incorporated are quite similar to each other. Did you check if the modelling results change substantially when running the calculations multiple times? Also, you use bulk soil water in different soil depths as available water sources. Contradictorily, these do not match with observed xylem values and you argue in the text that plants use soil water with differing isotopic composition. Additionally, measurement uncertainty in sources and plant xylem should be considered (see e.g. Kühnhammer et al. 2020). Having that said, I like that you compare rwu fractions derived with bulk soil samples and sampled roots as water sources. If you decide to keep the SIAR modelling, it would be interesting, in my opinion, to also look at tightly bound water as a potential set of sources and see how this changes the results. You could then also discuss the weaknesses of those purely statistical models and make use of your data to communicate potential issues with the usual approach (e.g. comparing bulk soil water and xylem water) to the scientific community.

**Reply:** We have deleted the calculation of plant water source contributions based on SIAR

modeling as suggested, but kept the conclusion that root water at 100-160 cm depths was the main

water source for the sampled plants. We believe that root water can reflect the water source of

trees better than bulk soil water (which has been more extensively used), for reasons detailed in

our response to comment 2.

7. As you sampled roots across soil depths, do you also have information on root length density

    across the profile or any other measure of root quantity across soil depths? I think this could

    add some interesting insights into the trees' water uptake strategies.

**Reply:** Unfortunately, we did not collect other root information such as root length density due to

the large workload.

*Technical comments:*

8. As you are discussing isotopic differences of soil water pools sampled as a reason for the

    observed mismatch to xylem water, you should pay close attention to specifying which soil

    water pool you are talking about. I suggest, either you always specify this or you clarify once

    that when talking about soil water in general you always refer to bulk soil water.

**Reply:** Thanks for your suggestions. We have identified the soil water we are referring to in the

revised manuscript.

9. In the copy I received for review, the figure quality seems too low, axis labels and names

    seem a bit blurry. It should be verified that all labels and names are easily readable.

**Reply:** The resolution of the figures was reduced by conversion from Word to pdf format. We have addressed to this issue in the revised manuscript. In addition, we have replotted the figures to improve their resolution.

10. Check for consistency of isotope terminology and avoid using (too many) different ones if it does not contribute to the readability of the text. You use all of the following: "isotopic composition", "isotope composition", "isotopes in water", "isotopic signature", "isotope signals", "isotopic signals", "isotope fingerprints", "H and O isotopes", "hydrogen and oxygen isotopes", "water isotopes", "isotopic values", "isotopic patterns"

    I know that sometimes it makes sense to mix it up a bit to avoid too many repetitions but I think you can eliminate at least a few of them :-D

    Also I would use the term "water stable isotopes" instead of "water isotopes"

**Reply:** Thanks for your suggestions. We have applied more consistent isotope terminology in the revised manuscript. In addition, we have changed "water isotopes" to "water stable isotopes" as suggested.

11. You use xylem to refer to measurements in the trunk. However, water within roots is also transported in the xylem (which you sampled to obtain the root isotopic values). I suggest to specify the use of words here or use trunk instead of xylem

**Reply:** We have changed "xylem water" to "trunk water" in the revised manuscript as suggested.

***Specific comments:***

12. Title: "insights into [an] isotopic mismatch between [bulk] soil water and *Salix matsudana* Koidz xylem water from root water isotope measurements"

**Reply:** According to the native English-speaking editor we invited, addition of 'an' seems inappropriate here. So, we have revised the title in the revised manuscript, as follows:

*"Insights into isotopic mismatch between bulk soil water and* Salix matsudana *Koidz trunk water from root water stable isotope measurements."*

13. Figure 2: I think the axis limits should be the same in all subplots (x-axis different for panel c).

**Reply:** We have standardized the x- and y-axis scales of the plots (see Figure 3 above) in the revised manuscript as suggested.

14. Figure 4 (caption): remove repetition of lc-excess

**Reply:** We have removed the repetition of lc-excess in the revised manuscript as suggested.

15. Figure 7: SD should also be displayed in the opposite direction, are the obtained distributions (of a RWU fraction at a certain depth) normally-distributed? If not, display uncertainty in a different way. Maybe it would make sense to display those distributions as boxplots

**Reply:** As mentioned in our reply to comment 6, we deleted Figure 7 due to the removal of references to the SIAR model.

16. Figure 8: I like the colour coding depicting the isotopic composition of the different compartments studied. Maybe this would come out even clearer if tree and soil background colours were a little more subtle (maybe grey scale?). Also see my general comment 1 to revaluate if you should emphasize fractionation as the main cause for the isotopic mismatch

**Reply:** We have replotted Figure 8 in the revised manuscript, as follows:

[Figure]

Figure 8 Schematic diagram of isotopic dynamics along the soil-root-trunk continuum. Color codes indicate isotopic composition of mobile water, bulk soil water and root water at indicated depths, groundwater and trunk water (from blue to brown representing low to high). The black asterisks indicate significant differences in the isotopic offset between root water and bulk soil water at the same depth ($p < 0.05$). The blue asterisks indicate significant differences in the isotopic offset between mobile water and bulk soil water at the same depth ($p < 0.05$).

17. Figure S2: Symbols are different sizes in different subplots

**Reply:** We have replotted Figure S2 and kept the same size for each subplot as suggested.

18. Figure S3: What is displayed? The figure legends and part of the caption states you display bulk water (BW) and tightly bound water (TW), but the caption also mentions mobile water (MW) – I assume that is a typo. I would also suggest changing the colours in Figure S3 to match with the colours in Figure 3, i.e. bulk soil water should always have the same colour

**Reply:** We have deleted Figure S3 and added the isotopic data in it to Figure 4 (see above).

19. Figure S4: the colours of the boxplots are hard to distinguish

**Reply:** We have deleted Figure S4 due to changes in the text.

20. Line 12: "at high temporal resolution" it reads as if all before mentioned parameters were sampled over time. However, this is not true for root xylem (only sampled on one occasion). Also, I think the perception of high temporal resolution is quite different depending on who you ask, especially with new in situ methods evolving. Maybe specify that you sampled twice a week/every X days?

**Reply:** We have clarified this as follows (Page 1 lines 13-16):

*"Thus, we measured the specific isotopic composition ($\delta^2H$ and $\delta^{18}O$) of each component (e.g., bulk soil water, mobile water, groundwater, trunk water and root water of* Salix matsudana *Koidz trees) with about three-day resolution in the soil-root-trunk continuum."*

21. Line 12-13: "to analyze isotopic dynamics in the soil-root-xylem continuum"

I don't really see a lot of focus on (temporal) isotope dynamics in your manuscript. You do not really discuss the variations over time (and only sampled roots once), maybe apart from the influence of GWC on soil water isotopes

**Reply:** We have revised this sentence in the revised manuscript, as follows (Page 1 lines 13-16):

*"Thus, we measured the specific isotopic composition ($\delta^2H$ and $\delta^{18}O$) of each component (e.g., bulk soil water, mobile water, groundwater, trunk water and root water of* Salix matsudana *Koidz trees) with about three-day resolution in the soil-root-trunk continuum."*

22. Line 17: I personally would not mention the TWW here again. "and plant fractionation" see my major point 1)

**Reply:** We have rephrased the Abstract in the revised manuscript (please see the main changes (4)) because of the changes in conclusions such as those regarding the TWW hypothesis and plant fractionation. In the revised manuscript, we have deleted this sentence.

23. Line 20: "isotopic offset occurred at the interface between the soil and *S. matsudana* roots" This statement is a bit misleading I think because if different soil water pools were not isotopically well mixed, the offset does not occurred at the interface between soil and roots but because plants only access a certain soil water pool

**Reply:** We have amended the results in the Abstract as follows (Page 1 lines 25-27):

*"In conclusion, the isotopic offset between bulk soil water and trunk water of* S. matsudana *reflected an isotopic mismatch between root water and bulk soil water associated with heterogeneity of the soil water."*

24. Line 28: "in [the] global hydrological cycle" or "in hydrological cycles", "terrestrial ecosystem[s]"

**Reply:** We have amended this sentence, as follows (Page 2 lines 36-39):

*"RWU also controls partitioning of infiltrated soil water between groundwater recharge and local atmospheric return through evapotranspiration (Knighton et al., 2020a; Knighton et al., 2020b), and thus plays a key role in the global hydrological cycle. In terrestrial ecosystems, plant transpiration accounts for more than 60% of total evapotranspiration and returns approximately 39% of incident precipitation to the atmosphere. "*

25. Line 42: exchange "in the movements" with "along the pathway"

**Reply:** We have deleted this sentence due to changes in the text.

26. Line 43: "but also [due] to ecohydrological separation (Brooks et al., 2010) [and] water

    isotope …"

**Reply:** We have deleted all contents regarding the TWW hypothesis (see our response to major

point 4).

27. Line 44-45: add reference to Chen et al. (2020) paper

**Reply:** We have added the reference in the revised manuscript as suggested.

28. Line 57: put "respectively" at the end of the sentence?

**Reply:** We have revised this sentence in the revised manuscript as suggested, as follows (Page 3

lines 71-73):

*"Poca et al. (2019) reported that arbuscular mycorrhizal fungi can enhance isotopic fractionation*

*during RWU, resulting in up to −24.6‰ and −2.9‰ differences in $\delta^2H$ and $\delta^{18}O$ values, between*

*soil and plant trunk water, respectively."*

29. Line 60: "incomplete extraction of water during cryogenic distillation could fractionate water

    isotopes" due to Rayleigh fractionation during the extraction process an incomplete

    extraction could not only fractionate water stable isotopes in the sample but surely does!

**Reply:** We have revised this sentence in the revised manuscript as suggested (Page 3 lines 76-78):

*"Incomplete extraction of water during cryogenic distillation fractionates water stable isotopes*

*(Gaj et al., 2017; Orlowski et al., 2018)."*

30. Line 61-62: "between [cryogenically extracted] stem water and source water"

**Reply:** We have adjusted the sentence as suggested (Page 3 lines 78-80):

*"Chen et al. (2020) found the common presence of significant isotopic deviations between cryogenically extracted trunk water and source water in nine woody plant species and demonstrated that this offset stems from methodological artifacts during cryogenic vacuum extraction."*

31. Line 63: "cryogenic extraction-associated methodological artifact" sounds a bit overly complicated

suggestion: "methodological artifacts during cryogenic vacuum extraction"

**Reply:** We have adjusted the sentence as suggested (Page 3 lines 78-80):

*"Chen et al. (2020) found the common presence of significant isotopic deviations between cryogenically extracted trunk water and source water in nine woody plant species and demonstrated that this offset stems from methodological artifacts during cryogenic vacuum extraction."*

32. Line 65: "specific process[es]"

**Reply:** We have amended this sentence, as follows (Page 3 lines 82-83):

*"Explanation of the isotopic offset between soil and trunk water is essential, but identifying roles of specific processes is generally hindered by the diversity of mechanisms that may be involved."*

33. Line 67: "along [the] soil-root-xylem continuum

**Reply:** We have adjusted the sentence as suggested (Page 3 lines 84-86):

*"Moreover, these mechanisms tend to have strongly interactive effects and may act on any compartment along the soil-root-trunk continuum such as soil matrix or soil-root interface or plant woody tissues."*

34. Line 68: "leading to the variation in water isotopes" I don't understand the statement of this subordinate clause.

**Reply:** We have rephrased this sentence and deleted the subordinate clause, as follows (Page 3 lines 84-86):

*"Moreover, these mechanisms tend to have strongly interactive effects and may act on any compartment along the soil-root-trunk continuum such as soil matrix or soil-root interface or plant woody tissues."*

35. Line 69: "roots preferentially use tightly bound water according to the TWW hypothesis"

In my opinion this statement is misleading as it might be attributed to the long-term availability of water in the soil. Mobile water (transported in big pores) percolates quickly below the plant rooting zone and therefore is only shortly plant available (during precipitation when rh is usually high and solar radiation low). Hence, roots do not prefer the more tightly bound water, they just use the water they have access to. I know however that it is termed like this in the TWW papers and different people might interpret this statement differently.

**Reply:** We have deleted all contents regarding the TWW hypothesis (see our response to major point 4).

36. Line 78-79: I would delete "during water transport from root to xylem" as you did not measure isotopes during the transport but at two locations (roots and trunk xylem)

**Reply:** We have deleted the clause in the revised manuscript as suggested.

37. Line 80-82: We hypothesize that 1) there is an isotopic deviation between xylem water of *S. matsudana* trees and their potential water sources, and that 2) this deviation might be due to a combination of multiple factors.

I think the second hypothesis is quite unspecific and that this is the case is already clear from your literature review. Be more precise on what you investigated and how your research contributes to disentangle combination of multiple contributing factors.

**Reply:** We have amended the sentence as suggested (Page 4 lines 101-103):

*"We hypothesize that mobile water is isotopically separate from bulk soil water in the soil matrix and isotopic deviation occurs between trunk water of* S. matsudana *trees and their potential water sources due to heterogeneity of the soil water."*

38. Line 98-99: We selected three sampling sites in the check-dammed channel of the Liudaogou catchment. Designated sites 1, 2 and 3 are located 50, 80 and 100 m upstream of the dam, respectively.

**Reply:** We have amended these lines as follows (Page 4 lines 120-121):

*"We selected three sampling sites in the check-dammed channel of the Liudaogou catchment, designated sites 1, 2 and 3, located 50, 80 and 100 m upstream of the dam, respectively."*

39. Line 100: "was chose for sampling tree" correct grammar

**Reply:** We have amended these lines as follows (Page 4 lines 121-123):

*"*Salix matsudana *Koidz, is one of the main tree species in the check-dammed catchment, so we chose* S. matsudana *as the sampling tree".*

40. Line 101: "includes" do you mean "consists of"?

**Reply:** Yes, we have amended the sentence as follows (Page 4 lines 123-125):

*"The soil at the site consists of sandy loam and loam according to the USDA classification system, with bulk density ranging from 1.4 to 1.6 g cm$^{-3}$."*

41. Line 102: "different soil [depths]" in soil science layers refer to differences in stratification, "in a sampling plot" which sampling plot? As you introduced the numbers before you could just specify

**Reply:** We have clarified this, as follows (Page 4 lines 125-126):

*"Water retention curves at 20, 30, 50, 100 and 150 cm soil depths at sampling site 1 are shown in Fig. S1."*

42. Line 104: "from the sampling plot[s]" or "sampling area"? or are you referring to a specific plot?

**Reply:** We have clarified this, as follows (Page 4 lines 126-127):

*"Meteorological data on precipitation and air temperature (with 30-min resolution) were obtained from a weather station located about 500 m from sampling site 1."*

43. Line 109: We collected root samples of [one] *S. matsudana* tree at each [of the three] sampling site[s]

**Reply:** We have clarified these points as follows (Page 5 lines 132-133):

*"We collected root samples from one* S. matsudana *tree and soil samples at selected soil depths (0-160 cm with 20 cm intervals) at each of the three sampling sites."*

44. Line 112-113: "was collected and measured its isotopic composition" correct grammar

**Reply:** We have corrected this sentence, as follows (Page 5 lines 136-138):

*"2-3 coarse roots (> 2 mm diameter) from each sub-cuboid were randomly selected and the roots from the top few centimeters of the topsoil were not artificially removed."*

45. Line 113: "collected disturbed soil samples at 0-160 cm depths" at what interval? Table one

suggests 10 cm increments

**Reply:** We have clarified this, as follows (Page 5 lines 144-145):

*"Moreover, we collected disturbed soil samples at 10 cm intervals from 0 to 100 cm depths and 20*

*cm intervals from 100-160 cm depths using a soil auger to measure soil particle size at sampling*

*site 1."*

46. Line 114-116: Did you measure particle size for both disturbed and undisturbed samples? Or

was this only done for disturbed samples and samples in cutting rings were used for water

retention curves?

**Reply:** We have clarified this, as follows (Page 5 lines 146-147):

*"We also collected undisturbed soil samples at 20, 30, 50, 100 and 150 cm depths using cutting*

*rings (100 cm$^3$ volume) to obtain water retention curves at the same sampling point."*

47. Line 119: "Our previous results have shown" This reads as it was referring to a previous

(published) study. Either you are missing a reference here or you should rephrase the

sentence to make clear that this refers to a previous campaign and is up to now unpublished

data

**Reply:** We have clarified this, as follows (Page 5 lines 152-153):

*"Previously unpublished data we obtained have shown that the isotopic composition of trunk*

*water of* S. matsudana *trees did not match bulk soil water in the dual-isotope space from May to*

*September 2018."*

48. Line 120-121: I am having difficulty to understand this sentence (referring to TWW).

**Reply:** We have deleted this sentence in the revised manuscript because it is related to the TWW hypothesis (see our response to major point 4).

49. Line 126: delete "For these analyses" otherwise it reads like precipitation samples are connected to beforementioned sample analysis

**Reply:** We have deleted "for these analyses" in the revised manuscript as suggested.

50. Line 124: "within the period when mobile water was available [ i.e. from August 4 to September 15 2019].

**Reply:** We have amended the sentence accordingly, as follows (Page 5 lines 157-159):

*"So, high frequency sampling (ca. 3-day temporal resolution) was applied to analyze the causes and locations of isotopic deviation during the period when mobile water was available (i.e. from August 4 to September 15 2019)."*

51. Line 128: do you know the depth of the groundwater table at the study site? At which depth did you sample groundwater?

**Reply:** We have added details about the groundwater table in the revised manuscript as suggested (Page 6 lines 163-164):

*"At our study site, the mean groundwater table depth was 3.6 m and groundwater samples were collected at ca. 30 cm depth from its surface."*

52. Line 129-131: How were soil samples (for isotopic analysis) stored? How was evaporation from samples prevented?

**Reply:** We have clarified these points as follows (Page 6 lines 166-167):

*"These soil samples were also rapidly placed in 10 mL vials that were sealed in the same manner as the root samples, then kept in a cool box until storage in the lab at −20 ℃."*

53. Line 138: "placed in 10 mL vials and wrapped in parafilm" I guess you sealed the vials with caps and then secured the caps with parafilm? Parafilm is not 100% gas tight but permeable to water vapor.

**Reply:** We have clarified this, as follows (Page 6 lines 177-179):

*"Pieces of the de-barked and de-leaved twigs, 30 mm long, were then immediately placed in 10 mL vials, the vials were sealed with caps then the caps were secured with Parafilm. These samples were also kept in a cool box until storage in the lab at 4℃."*

54. Line 141-142: "Similarly, of 30 mm long pieces of the de-barked twigs were immediately placed in 10 mL vials and wrapped in parafilm." Is this a repetition or did you want to specify the sampling in different tree heights? Maybe streamline to avoid redundancies

**Reply:** This sentence describes the collection of trunk water at different sampling heights. We have revised these sentences to make them clearer, as suggested (Page 6 lines 173-179).

*"Tree samples were collected simultaneously with the soil samples. These consisted of twigs collected from the south-facing side of three* S. matsudana *trees at 250 cm height on each sampling occasion. In addition, samples of trunk at selected tree heights (150, 250, 350, 450 cm) were collected on August 18, 2019. Bark and phloem were peeled from fully suberized branches to avoid perturbance of trunk water isotopic composition by fractionation. Pieces of the de-barked and de-leaved twigs, 30 mm long, were then immediately placed in 10 mL vials, the vials were sealed with caps then the caps were secured with Parafilm. These samples were also kept in a cool box until storage in the lab at 4℃."*

55. Line 163: b does not appear in your lc-excess formula (is essentially the lc-excess)

**Reply:** We thank the reviewer for alerting us to this error and we have revised the lc-excess

formula in the revised manuscript, as follows (Page 7 line 204):

"$lc - excess = \delta^2 H_s - a\delta^{18}O_s - b$        "

56.  Line 166-167: In my opinion you do not need the link to the TWW here

**Reply:** We have deleted this sentence because of the removal of references to TWW hypothesis (see

our response to major point 4).

57.  Line 174: "To compare the isotopic composition of root and soil water at the same depth"

    Actually you only indirectly compare root and bulk soil water with this. If you aim at

    comparing those two, I think there is better, different approaches. I would rather say you

    compare their contribution to xylem water. See also my major comment 5) here.

**Reply:** We have deleted this sentence because of the removal of references to SIAR model (see

our response to major point 6).

58.  Line 177: "Based on results of the soil water and root water isotope analysis"

I do not get what the decision criteria was here. Please specify.

**Reply:** We have deleted this sentence because of the removal of references to SIAR model (see

our response to major point 6).

59.  Line 178: "soil [and root] water sources were divided into …"

Then you can delete the next sentence

**Reply:** We have deleted this sentence because of the removal of references to SIAR model (see

our response to major point 6).

60. Line 181: "because plant water use does not generally cause fractionation of hydrogen and

oxygen isotopes"

This contradicts with the statement in your paper that fractionation during rwu influences the

xylem isotope values. I would rephrase and write: "assuming no fractionation during plant

water uptake"

**Reply:** We have deleted this sentence because of the removal of references to SIAR model (see

our response to major point 6).

61. Line 202: "overlapped with bulk soil water (Fig 2d)" Actually one needs to look at Figure 2a

and d to see the overlap

**Reply:** We have clarified this in the revised manuscript, as suggested.

62. Line 208: unit ‰ is missing

**Reply:** The information expressed in this sentence is limited, thus we deleted it, but we have

added another table (see Table 2 below), showing the water stable isotopes and lc-excess values

for all water samples.

Table 2 Water stable isotopes (see Fig. 3) and lc-excess values (Fig. 4) for all water samples. Range
values show min, max (mean).

| Water samples | N | $\delta^2$H range (‰) | $\delta^{18}$O range (‰) | lc-excess range (‰) |
|---|---|---|---|---|
| Groundwater | 22 | −64.7, −63.2 (−64.1) | −9.1, −8.6 (−8.8) | −3.2, −1.0 (−2.4) |
| Mobile water | 191 | −71.7, −48.8 (−61.9) | −10.7, −6.9 (−8.7) | −5.7, 4.6 (−1.2) |
| Bulk soil water | 203 | −89.5, −38.1 (−64.5) | −11.9, −5.1 (−8.3) | −12.5, −1.7(−6.7) |
| Less mobile water | 176 | −99.9, −24.6 (−65.1) | −11.2, −2.4 (−8.0) | −23.9, −2.8 (−9.9) |
| Root water | 156 | −71.3, −43.9 (−63.3) | −8.9, −6.5 (−7.6) | −16.9, −2.1 (−10.7) |
| Trunk water | 61 | −70.4, −62.8 (−66.7) | −8.4, −7.3 (−7.7) | −17.1, −9.0 (−13.5) |

63. Line 211: I would not mention TWW here, as it is part of the discussion.

**Reply:** We have deleted this sentence because of the removal of references to TWW hypothesis (see our response to major point 4).

64. Line 212-213: "suggesting that xylem water was isolated from all potential water sources."

    Actually, it only means that xylem water did not reflect bulk soil water sources and not that it is independent from all potential water sources. Also contradicts with the next sentence and is strictly speaking already discussing the results.

**Reply:** We have deleted this sentence, but added a brief discussion about this issue in Section 4.1.2 of the Discussion "Isotopic offset between bulk soil water and root water", as follows (Page 13 lines 398-406):

*"Overall, the most plausible explanation for isotopic mismatch between root water and bulk soil water in dual-isotope plots is that bulk soil water is not representative of available plant water sources because of the heterogeneity of bulk soil water. As shown in Fig. 3, less mobile water overlapped isotopically with root water after removing the influence of mobile water. The rapidity of mobile water's passage through soil reduces its contact with mineral surfaces, and hence its nutrient concentrations (McDonnell, 2017; Sprenger et al., 2019). Thus, plants may have used large amounts of less mobile water that was strongly affected by evaporative effects in the presented study, isotopically distinct from mobile water and groundwater, and with similar isotopic composition to trunk water."*

McDonnell, J.J.: Beyond the water balance, Nat. Geosci., 10, 396-396, 2017.

Sprenger, M., Llorens, P., Cayuela, C., Gallart, F., and Latron, J.: Mechanisms of consistently

disjunct soil water pools over (pore) space and time. Hydrol. Earth Syst. Sci., 23, 2751-2762,

2019.

65.  Line 213-214: streamline sentence to avoid repetition of tightly bound water, also includes

discussion of results already

**Reply:** We have deleted this sentence because it is more suitable for the *Discussion* than the

*Results*.

66.  Line 232: "[horizontal] distance"

**Reply:** We have amended the sentence as suggested (Page 10 lines 291-292):

*"As shown in Fig. 6b and d, there were no significant differences (p > 0.05) in isotopic*

*composition ($\delta^2H$ and $\delta^{18}O$) of either root water or bulk soil water between 40 cm and 80 cm*

*horizontal distance from selected tree trunks."*

67.  Line 233: exchange "> 80 cm" with "within 80 cm"

**Reply:** We will amend the sentence as suggested (Page 10 lines 291-293):

*"As shown in Fig. 6b and d, there were no significant differences (p > 0.05) in isotopic*

*composition ($\delta^2H$ and $\delta^{18}O$) of either root water or bulk soil water between 40 cm and 80 cm*

*horizontal distance from selected tree trunks, suggesting that isotopic composition of the bulk soil*

*water was horizontally homogenous within 80 cm from tap roots."*

68.  Line 239 & 241: $\delta^2O$ should be $\delta^{18}O$ I assume, also stay consistent with the subscripts, here

the subscript soil refers to bulk soil water but subscript BW also exists

**Reply:** We thank the reviewer for alerting us to this error and have revised this sentence in the

revised manuscript as follows (Page 10 lines 302-303):

*"Similarly, a strong correlation was observed between $\Delta^{18}O$ ($\Delta^{18}O = \delta^{18}O_{soil} - \delta^{18}O_{trunk}$) and $\Delta^2H$*

*($\Delta^2H = \delta^2H_{soil} - \delta^2H_{trunk}$) soil-trunk offsets during August 4 to September 15."*

69. Line 243: "These results show that water isotopes, especially hydrogen isotopes, changed

    between…"

    Water stable isotopes do not change, the ratios of the isotopes change.

**Reply:** We have deleted this sentence because of the changes in content.

70. Line 244: "supporting our first hypothesis" belongs into discussion.

**Reply:** We have deleted this sentence in the revised manuscript because it is related to the TWW

hypothesis (see our response to major point 4).

71. Line 246: Specify heading of subsection, contribution of what?

**Reply:** We have deleted Section 3.4 of the Results, "Contributions", because of the removal of

references to the SIAR model (see our response to major point 6).

72. Line 248: "Potential sources of plant xylem water were determined using…"

    you do not determine potential sources but the contribution of these sources to rwu.

**Reply:** We have deleted this sentence because of the removal of references to SIAR model (see

our response to major point 6).

73. Line 256: "Separation of mobile water and bulk soil water in the soil matrix"

    Your manuscript deals with this separation a lot and it is also a big part of your study design.

    Why is this not one of your hypothesis/aims?

**Reply:** Thanks for your suggestion. We have changed it to one of our hypotheses in the revised

manuscript, as follows (Page 4 lines 101-103):

*"We hypothesize that mobile water is isotopically separate from bulk soil water in the soil matrix and isotopic deviation occurs between trunk water of* S. matsudana *trees and their potential water sources due to heterogeneity of the soil water."*

74. Line 257: exchange "covered" with "experimental"

**Reply:** We have revised this sentence in the revised manuscript as suggested (Page 11 lines 320-321):

*"At our study site during the experimental period (August 4 to September 15, 2019), a clear isotopic separation between mobile and bulk soil water was observed."*

75. Line 261: "clear [isotopic] separation"

**Reply:** We have added "isotopic", as suggested (Page 11 lines 340-342):

*"The lc-excess values of mobile and bulk soil water consistently differed significantly, although GWC varied greatly, suggesting a clear isotopic separation between mobile and bulk soil water that is not affected by GWC."*

76. Line 266-268: Reference for statement missing

**Reply:** We have added references in the revised manuscript, as follows (Page 11 lines 322-329):

*"Gierke et al. (2016) examined the stable isotopic composition of precipitation, bulk soil water and trunk water in a high elevation watershed and their results suggested that mobile water was primarily associated with summer thunderstorms, and thus subject to minimal evaporative loss. In contrast, less mobile water was derived from snowmelt, filling small pores in the shallow soils. Allen et al. (2019) characterized the occurrence of winter and summer precipitation in plant trunk samples using a seasonal origin index and found that winter precipitation was the predominant water source for midsummer transpiration in sampled beech and oak trees. Due to seasonal*

*isotopic cycles in precipitation, there may be clear distinctions in the isotopic composition of*

*mobile water and less mobile water derived from precipitation falling at different times (Bowen et*

*al., 2019)."*

Allen, S.T., Kirchner, J.W., Braun, S., Siegwolf, R.T.W., and Goldsmith, G.R.: Seasonal origins of

soil water used by trees, Hydrol. Earth Syst. Sci., 23, 1199-1210, 2019.

Bowen, G.J., Cai, Z.Y., Fiorella, R.P., and Putman, A.L.: Isotopes in the water cycle: regional- to

global-scale patterns and applications, Annu. Rev. Earth Planet. Sci., 47, 453-479, 2019.

Gierke, C., Newton, B.T., and Phillips, F.M.: Soil-water dynamics and tree water uptake in the

Sacramento Mountains of New Mexico (USA): a stable isotope study, Hydrogeol. J., 24, 805-

818, 2016.

77. Line 268-269: "Due to the seasonal variation in precipitation, winter and summer

precipitation have different isotope signals"

Both half sentences state the same thing. Seasonal variation is not the cause for isotopic

differences between winter and summer precipitation

**Reply:** Following the suggestion we have amended this sentence, as follows (Page 11 lines 327-329):

*"Due to seasonal isotopic cycles in precipitation, there may be clear distinctions in the isotopic*

*composition of mobile water and less mobile water derived from precipitation falling at different*

*times (Bowen et al., 2019)."*

78. Line 272: place "at our study site" at the beginning of the sentence

**Reply:** We have amended the sentence as suggested (Page 11 lines 329-330):

*"At our study site, precipitation in winter (December-February) and summer (June-September)*

*accounted for 2% and 77% of total average annual precipitation (464 mm) from 2003 to 2019,*

*respectively."*

79. Line 274: "caused by other factors, and not necessarily by seasonal variation in precipitation"

   Observed differences could also stem from isotopic differences of individual precipitation

   events. Did you check if with high and low intensities are systematically different? If so, high

   intensity precipitation events might percolate faster into the soil and also contribute bigger

   ´water quantities to sampled mobile water.

**Reply:** Thanks for your suggestions. In the revised manuscript we have changed our discussion of

this issue based on the antecedent precipitation events, as follows (Page 11 lines 331-337):

*"Notably, there was a major rainstorm the day before the sampling (August 3), with 63 mm*

*precipitation. The mean GWC in 0-50 cm and 100-150 cm layers reached 17.4 ± 2.7% and 10.8 ±*

*1.5% between August 4 and August 7, respectively. These results imply that precipitation greatly*

*supplemented the water in the upper soil layer. So mobile water collected by suction lysimeters*

*during this period contained a considerable proportion of water from the rain event on August 3.*

*In contrast, bulk soil water contained not only mobile water from this rain event, but also*

*antecedent less mobile water that could not be extracted by a suction lysimeter, resulting in the*

*isotopic separation between mobile water and bulk soil water."*

80. Line 277: "with [increasing] soil depth"

**Reply:** We have added "increasing" as suggested (Page 12 lines 356-357):

*"The effect of soil evaporation on bulk soil water gradually weakens with increasing soil depth."*

81. Line 281-282: "Although the mixing of mobile and tightly bound water conflicts with the

original hypothesis of Brooks et al. (2010)"

Does it really disagree? Or is it more a question of the degree of mixing or rather that mixing

between soil water pools is not complete as previously assumed?

**Reply:** We have deleted this sentence because of the removal of references to TWW hypothesis

(see our response to major point 4).

82. Line 282-285: Increased mixing with soil depth was e.g. also observed/mentioned by

Sprenger et al. (2016) and Kübert et al. (2020)

https://agupubs.onlinelibrary.wiley.com/doi/full/10.1002/2015RG000515

https://www.frontiersin.org/articles/10.3389/fpls.2020.00387/full

**Reply:** Thanks for this suggestion. We have added these references in the revised manuscript, as

follows (Page 12 lines 361-362):

*"Both mobile water and less mobile water in deep layers are more fully mixed than in shallow*

*layers (Sprenger et al., 2016; Kubert et al., 2020)."*

83. Line 290-291: "driven by spatiotemporal dynamics of soil water profiles associated with soil

evaporation"

What do you mean with that? The temporal (and spatial) differences in infiltration and

evaporation and how those influence the sampled soil water sources?

**Reply:** We have deleted this sentence because of the removal of references to TWW hypothesis

(see our response to major point 4).

84. Line 296-297: "These results showed that the isotopic offset between plant root water and

soil water occurred at the root-soil interface."

I think this statement is misleading. I think this statement is misleading. In my opinion it points towards the explanation that bulk soil water is not representative of available plant water sources (see next paragraph). This heterogeneity does however not only apply to the root-soil interface. Or am I missing something here?

**Reply:** As mentioned in our response to comment 2, we have revised this sentence in the revised manuscript as follows (Page 13 lines 411-413):

*"However, these findings are not consistent with the greater $^2H$ enrichment in root water than in bulk soil water (differences up to 8.6‰) we detected, suggesting that soil-root isotopic offsets are more likely to be caused by the complexity of root systems and heterogeneity of bulk soil water than isotopic fractionation during root water uptake."*

85. Line 302: "that is strongly affected by evaporative effects [in the presented study], isotopically separated from mobile water and groundwater and shows similar enriched isotopic signals [than xylem water]"

**Reply:** We have amended the sentence as suggested (Page 13 lines 403-406):

*"Thus, plants may have used large amounts of less mobile water that was strongly affected by evaporative effects in the presented study, isotopically distinct from mobile water and groundwater, and with similar isotopic composition to trunk water."*

86. Line 303-304: "This hypothesis is corroborated by the overlap in isotopic composition between root and bulk soil water at 0-60 cm depths (Fig. 2 and 5)."

    Why? As I understand it, this speaks against the statement in the sentence before.

**Reply:** We have deleted this sentence in the revised manuscript because it is related to the TWW hypothesis (see our response to major point 4).

87. Line 304-309: "We considered whether bulk soil water isotopes can represent isotopic values of tightly bound water used by plants. Generally, the water designated 'bulk soil water' includes mobile and tightly bound water due to limitations of water extraction technology when assessing the TWW hypothesis. Thus, the proportion of mobile water in the bulk soil water increases as soil moisture increases, resulting in isotopic deviation between root water and bulk soil water."

I do not understand the argument here. Maybe rephrasing helps :-D

**Reply:** We have deleted this sentence in the revised manuscript because it is related to the TWW hypothesis (see our response to major point 4).

88. Line 331-332: "Under the assumption that plant fractionation does not occur"

Do you mean that the isotopic composition of water does not change during within plant transport? Fractionation is mainly believed to occur during rwu I think. However, as you sampled the roots, you eliminate this influencing factor.

**Reply:** We have deleted this sentence in the revised manuscript because of the changes in content.

89. Line 341: "also has residence times in branches and roots"

What do you mean with this? How is it different from the time lag due to transport from roots to branches? Do you refer to an influence of xylem water storage?

**Reply:** We have clarified this, as follows (Page 14 lines 442-444):

"*As the time required for isotopic tracer ($D_2O$) to move from the base of a trunk to the upper crown of a tree reportedly ranges from 2.5 to 21 days (Meinzer et al., 2016), the isotopic*

*composition of trunk water may differ from that root water collected on the same day (August*

*18).*"

Meinzer, F.C., Woodruff, D.R., Marias, D.E., Smith, D.D., Mcculloh, K.A., Howard, A.R., and

Magedman, A.L.: Mapping 'hydroscapes' along the iso- to anisohydric continuum of stomatal

regulation of plant water status, Ecol. Lett., 19, 1343-1352, 2016.

90. Line 346: "that isotope enrichment may have been present in the unsampled branches"

    Why would you measure a potential enrichment caused by unsampled branches by sampling

    at different heights? What do you mean with enrichment present in unsampled branches?

**Reply:** We have revised this sentence in the revised manuscript as follows (Pages 14-15 lines 446-

449):

*"Moreover, to test the possibility that isotopic composition of trunk water may be heterogeneous*

*at different tree heights, we collected trunk water at 150-450 cm tree heights on August 18, 2019,*

*and found no significant differences (p > 0.05) (Fig. S3)."*

91. Line 348: "xylem water was [isotopically] more enriched than…"

**Reply:** We have amended the sentence as follows (Page 15 lines 460-462):

*"Furthermore, previous studies have provided indications that trunk water becomes more*

*enriched in $^{18}O$ due to the temporal declines in sap flow rates (Martin-Gomez et al., 2017) and the*

*mixture of trunk water with leaf water (Brandes et al., 2007)."*

Brandes, E., Wenninger, J., Koeniger, P., Schindler, D., Rennenberg, H., Leibundgut, C., Mayer, H.,

and Gessler, A.: Assessing environmental and physiological controls over water relations in a

Scots pine (*Pinus sylvestris* L.) stand through analyses of stable isotope composition of water

and organic matter. Plant Cell Environ., 30,113-127, 2007.

Martin-Gomez, P., Serrano, L., and Ferrio, J.P.: Short-term dynamics of evaporative enrichment of

xylem water in woody stems: implications for ecohydrology, Tree Physiol., 37, 511-522, 2017.

92.   Line 352-353: "However, we found that the xylem water contained [more of the depleted]

isotopic signal of deep roots than [of the] enriched signal from shallow roots. The results

show that there was no isotopic fractionation during water transport from root to xylem"

I am not sure if this allows the conclusion that no fractionation during transport took place.

However, I agree that it strongly suggests it. I would also specify here that you did not

observe an enrichment (fractionation could go both ways) during transport as other authors

suggested.

**Reply:** We have amended the sentence as follows (Page 15 lines 460-464):

*"Furthermore, previous studies have provided indications that trunk water becomes more*

*enriched in $^{18}O$ due to the temporal declines in sap flow rates (Martin-Gomez et al., 2017) and the*

*mixture of trunk water with leaf water (Brandes et al., 2007). However, we did not find that trunk*

*water of the trees we sampled had higher $^{18}O$ values than root water (Fig. 3). Therefore, in this*

*study, root water partially overlapped with trunk water isotopic composition, we believe it reflects*

*the selective utilization of water source rather than isotopic fractionation within woody tissues."*

Brandes, E., Wenninger, J., Koeniger, P., Schindler, D., Rennenberg, H., Leibundgut, C., Mayer, H.,

and Gessler, A.: Assessing environmental and physiological controls over water relations in a

Scots pine (*Pinus sylvestris* L.) stand through analyses of stable isotope composition of water and organic matter. Plant Cell Environ., 30,113-127, 2007.

Martin-Gomez, P., Serrano, L., and Ferrio, J.P.: Short-term dynamics of evaporative enrichment of xylem water in woody stems: implications for ecohydrology, Tree Physiol., 37, 511-522, 2017.

93.   Line 358: exchange "covered" with "experimental" or write "period covered"

**Reply:** We have revised this sentence in the revised manuscript as suggested (Page 16 lines 484-485):

*"At our study site during the experimental period, the isotopic offset existed between trunk water of* S. matsudana *trees and bulk soil water."*

94.   Line 359-360: "isotopic offset [exists] between xylem water and [bulk soil water]"

As you elaborate in your manuscript, bulk soil water might not reflect all available soil water sources.

**Reply:** We have revised this sentence, as follows (Page 16 lines 495-496):

*"These findings suggest that the isotopic offset between bulk soil water and trunk water was due to isotopic mismatch between root water and bulk soil water associated with heterogeneity of the soil water."*

95.   Line 361: "and water flow paths" what do you mean with that? Infiltration along preferential flow paths?

**Reply:** We have clarified this, as follows (Page 16 lines 489-490):

*"In the soil matrix, bulk soil water generally had lower lc-excess values than mobile water, due to*

*effects of soil evaporation and mixture of newly infiltrated mobile and less mobile water with*

*increasing depth."*

96.  Line 369: "the [estimated] contribution of roots in these depths to xylem water [was] 74%."

**Reply:** We have deleted this sentence because of the removal of references to SIAR model (see

our response to major point 6).

**Point-by-point by responses to Reviewer 2's comments**

*Major points:*

1.  In the manuscript entitled, "Insights into isotopic mismatch between soil water and *Salix matsudana* Koidz xylem water from root water isotope measurements", results from a field study show isotopic differences between different soil water and plant water pools. The dataset is interesting and results from intensive measurements. I think some of the findings are potentially interesting and warrant publication. However, the current presentation has some weaknesses and ambiguities that need to be addressed. Ultimately, the imprecise use of jargon obscures the interpretation and implications of the study.

**Reply:** We thank the reviewer for the constructive comments and suggestions, we have carefully considered them and tried our best to reduce the highlighted weaknesses and ambiguities in the manuscript, especially the problem about the "two water worlds" hypothesis.

2.  Throughout, references are made to ecohydrological separation and the two-water-worlds (TWW) hypothesis. However, I am not sure how the authors define these phenomena; their definitions seem different from my own, different from the literature that they are citing, and they may change throughout. At times, it seems that these terms just mean "there are isotopic differences", which is not particularly novel to identify. The paper would greatly benefit from less use of jargon. It should be more explicitly stated what is being tested. From my reading of this, the key questions of this paper are "Does root water isotopic composition match that of soils' bulk, mobile, and bound water fractions of soils at the same depths". Then there is a second question "How does choice of stem sample (from four heights) or choice of soil versus roots influence inferences of water uptake depths". Neither of these questions is

especially related to TWW or ecohydrological separation. Frankly, the duration of the study is too short to assess either TWW or ecohydrological separation because any observed differences between plant water and groundwater isotope ratios might be a product of lags, rather than the use of fundamentally different sources. The process Brooks et al referred to as ecohydrological separation was only observable through using measurements across multiple seasons.

**Reply:** We agree that the short experimental period and the focus on phenomena that are not directly related to the "two water worlds" (TWW) hypothesis hinder concise, meaningful discussion of the hypothesis and ecohydrological separation. Reviewer 1 also suggested that references to the TWW hypothesis should be reduced. Therefore, we have deleted all contents about the TWW hypothesis and paid more attention to the soil water's heterogeneity through the comparison of mobile water, bulk soil water, and derived characteristics of less mobile water at the same depths, and the impact of this heterogeneity on plant water uptake in the revised manuscript.

3. The isotopic differences between root water and soil water is key to the conclusions made in this paper. The authors seem to suggest that the water in roots should match the water in soils around them. They did not, and this was interpreted as potential fractionation. However, roots can transport water from different locations. I would only expect similarity between roots and surrounding soils if fine roots were sampled. For coarser roots, as used in this study, I would expect those roots to transport water from much deeper depths and integrate large volumes of soil water. Thus, it is not clear that "a combination of plant fractionation and TWW-type separation" (which, again, needs to be clarified) is needed to explain the observations here. This needs to be further discussed. Potentially, additional excavations may be warranted to

identify whether the size of root samples could include fine roots that extend substantially

deeper.

**Reply:** Thanks for your suggestions. We have changed the conclusion that isotopic fractionation

leads to the observed mismatch between root water and bulk soil water at the same depth in the

manuscript, for two reasons. First, the water in the sampled coarse roots (> 2 mm diameter) does

not necessarily match the bulk soil water around them because sampled coarse roots can transport

and mix water from different locations, as suggested. Second, as suggested by Reviewer 1, recent

studies on isotopic fractionation have found stronger $^2$H depletion in trunk water/root water than in

bulk soil water (e.g., Poca et al., 2019; Vargas et al., 2017). However, these findings are not

consistent with our finding that root water had higher $\delta^2$H values than bulk soil water (up to

8.6‰). Most importantly, we found that the isotopic composition of root water deviated from that

of bulk soil water, but overlapped with the values derived for less mobile water (see Figure 3

below). Thus, we concluded that soil-root isotopic offsets are more likely to be caused by the

complexity of root systems and the heterogeneity of bulk soil water than isotopic fractionation

during root water uptake. Hence, we have added the following discussion regarding this issue in

the revised manuscript (Pages 12-13 lines 375-413):

[revised manuscript text omitted]

2017.

***Specific comments:***

4. 14, 18, 77: I think the author is referring to ecohydrological separation, not the "two water

worlds hypothesis", which is about streamflow. However, I am generally unclear on this.

**Reply:** We agree that the short experimental period and the focus on phenomena that are not

directly related to the "two water worlds" (TWW) hypothesis hinder concise, meaningful

discussion of the hypothesis and ecohydrological separation (see our response to major point 2).

Thus, we have deleted all contents regarding the TWW hypothesis and ecohydrological separation.

5.    20: From the abstract, it is not clear why this is "in conclusion"

**Reply:** We have rephrased the abstract, please see the main changes (4).

6.    21-23: It is better to say what that contribution is, and what those insights are, rather than just mentioning that they exist.

**Reply:** We have revised this sentence as suggested, as follows (Page 1 lines 27-29):

*"Our results illuminate relationships between the isotopic composition of soil water of various mobility, root water and trunk water that may be useful for advancing our understanding and representation of root water uptake and transport."*

7.    41-44 If movement alone creates the change, I'd argue that this statement violates laws of mass conservation. Any changes must be matched by equal and opposite changes elsewhere. Ecohydrological separation is not a change from soil to root to xylem.

**Reply:** We thank the reviewer for alerting us to this error and have deleted this sentence in the revised manuscript.

8.    46-48 See comments above. Please re-read Brooks et al.

**Reply:** We have deleted all contents regarding the "two water worlds" hypothesis in the revised manuscript as suggested (see our response to major point 2). We will also read papers concerning the "two water worlds" hypothesis more carefully in the future.

9.    48-49 How would the hypothesis be supported by groundwater and streams? This does not make sense to me. The hypothesis is related to plant-available soil water.

**Reply:** We have deleted this sentence in the revised manuscript because it is related to the "two water worlds" hypothesis.

10. 51: What does "related to infiltration" mean? Is "tightly bound" water not related to infiltration?

**Reply:** We have deleted this sentence in the revised manuscript because it is related to the "two water worlds" hypothesis.

11. 120-124: This is a short period for assessing ecohydrological separation, especially in a dry-climate region. Identifying ecohydrological separation phenomena requires detecting bypassing of stored waters.

**Reply:** We agree that the short experimental period hinders concise, meaningful discussion of ecohydrological separation. Therefore, we have deleted all contents regarding the "two water worlds" hypothesis in the revised manuscript as suggested (see our response to major point 2).

12. 180-182 Given the interest in fractionation upon uptake, why not set it to a value more consistent with others in the literature.

**Reply:** Thanks for your suggestions. We have changed the conclusion that the observed isotopic mismatch between root water and bulk soil water at the same depth was caused by isotopic fractionation, as advised by both reviewers (please see our response to main point 3). Moreover, application of the SIAR model does not strengthen the story, as pointed out by Reviewer #1, so we have deleted the calculation of plant water source contributions based on the SIAR model.

13. 193-205 I do not understand why slopes in dual-isotope space are being used here. I don't think they are the most effective way to make the comparisons that are being made. First, were these slopes fit orthogonally? They should be. Second, when one line is compared to

another, is that the result of an ANCOVA test? Third, are the p values the fits of the lines, or p

values for the comparisons being described? I do not understand why fitted lines, rather than

the actual values, are being compared; it seems that the interpretations relate to the actual

values.

**Reply:** We have used actual values instead of the slopes for analysis in accordance with this

comment in the revised manuscript, as follows (Page 8 lines 236-249):

*"Stable isotopic composition ($\delta^2H$ and $\delta^{18}O$) of all water samples are shown in Fig. 3a and Table*

*2. The slope and intercept of the local meteoric water line (LMWL, $\delta^2H = 7.67 \, \delta^{18}O + 5.91$, $R^2 =$*

*0.96) were lower than those of the global meteoric water line (GMWL, $\delta^2H = 8 \, \delta^{18}O + 10$) (Craig,*

*1961). Mobile water at all depths (i.e. 20, 30, 50, 100 and 150 cm) typically fell on the LMWL and*

*groundwater was isotopically similar to mobile water at 150 cm depth (Fig. 3b). Bulk soil water*

*partly overlapped isotopically with mobile water but it generally plotted below mobile water (Fig.*

*3a and c). Less mobile water deviated from the LMWL and overlapped with root water and trunk*

*water (Fig. 3a and d). Trunk water was isotopically similar to root water at 100-160 cm depths*

*(Fig. 3a and e-f)."*

We have added another table (see Table 2 below), showing the water stable isotopes and lc-excess

values for all water samples.

Table 2 Water stable isotopes (see Fig. 3) and lc-excess values (Fig. 4) for all water samples.

Range values show min, max (mean).

| Water samples | N | $\delta^2H$ range (‰) | $\delta^{18}O$ range (‰) | lc-excess range (‰) |
|---|---|---|---|---|
| Groundwater | 22 | −64.7, −63.2 (−64.1) | −9.1, −8.6 (−8.8) | −3.2, −1.0 (−2.4) |
| Mobile water | 191 | −71.7, −48.8 (−61.9) | −10.7, −6.9 (−8.7) | −5.7, 4.6 (−1.2) |

| | | | | |
|---|---|---|---|---|
| Bulk soil water | 203 | −89.5, −38.1 (−64.5) | −11.9, −5.1(−8.3) | −12.5, −1.7(−6.7) |
| Less mobile water | 176 | −99.9, −24.6 (−65.1) | −11.2, −2.4 (−8.0) | −23.9, −2.8 (−9.9) |
| Root water | 156 | −71.3, −43.9 (−63.3) | −8.9, −6.5 (−7.6) | −16.9, −2.1 (−10.7) |
| Trunk water | 61 | −70.4, −62.8 (−66.7) | −8.4, −7.3 (−7.7) | −17.1, −9.0 (−13.5) |

14. 208 What are these uncertainties: standard deviations? All depths are lumped together for soil water?

**Reply:** These numbers are in the form of "mean ± standard deviation". However, the information expressed in this sentence is limited, thus we deleted it, but added another table (see Table 2 above), showing the water stable isotopes and lc-excess values for all water samples.

15. 211 Ecohydrological separation was a process that could only be revealed from a long duration of sampling. How can this short period of measurement show ecohydrological separation?

**Reply:** Thanks for your suggestion. We agree that the short experimental period hinders concise, meaningful discussion of ecohydrological separation. Therefore, we deleted all contents about ecohydrological separation and paid more attention to the soil water's heterogeneity through the comparison of mobile water, bulk soil water, and derived characteristics of less mobile water at the same depths, and the impact of this heterogeneity on plant water uptake in the revised manuscript.

16. 212 What is the statistical test used?

**Reply:** It was the Tukey-Kramer HSD test, as we have stated in the revised manuscript (Page 9 lines 256-259).

*"As shown in Fig. 4 and Table 2, the mean lc-excess values of groundwater and mobile water did not significantly differ ($p > 0.05$), and they were significantly higher than those of bulk soil water,*

*less mobile water and trunk water (Tukey-Kramer HSD, p < 0.05) during the sampling period*

*(August 4 to September 15, 2019)."*

17. 217 Does "always" mean every single value was different? I do not understand how one
would test that.

**Reply:** The intended meaning is that at different sampling depths (i.e. 20 cm, 30 cm, 50 cm, 100

cm, 150 cm), there were significant differences between mobile water and bulk soil water lcexcess (see Figure 4A-F below). To clarify this, we have modified the sentence in the revised

manuscript as follows (Page 9 lines 272-274):

*"At every sampling depth, the mean lc-excess of mobile water was always higher than that of bulk*

*soil water and less mobile water (Tukey-Kramer HSD, p < 0.05) during the whole sampling*

*period (Fig. 4A-E)"*

[Figure]

Figure 4 (a-f) Temporal dynamics of hydrological conditions (precipitation and gravimetric water content, GWC) and lc-excess values (these values are means and standard deviations for three sites) of groundwater (GW), trunk water (TW), mobile water (MW), less mobile water (LMW) and bulk soil water (BW) at indicated depths (20, 30, 50, 100 and 150 cm) during the period August 3 to September 15, 2019. (A) Boxplots of total MW (N=191), GW (N=22), BW (N=204), TW (N=61) and LMW (N=176) lc-excess values. (B-F) Boxplots of MW and BW at 20 cm (MW,

N=40; BW, N=42; LMW, N=39), 30 cm (MW, N=40; BW, N=40; LMW, N=34), 50 cm (MW, N=38; BW, N=40; LMW, N=33), 100 cm (MW, N=36; BW, N=40; LMW, N=34) and 150 cm (MW, N=37; BW, N=42; LMW, N=36) depths. The top and bottom of each box are the 25th and 75th percentiles of the samples, respectively. The black line in each box is the sample median. Trunk water and potential water sources that do not share a letter are significantly different ($p <$ 0.05, Tukey-Kramer HSD).

18. Section 3.3 Are these fine roots only? If not, I see no reason to think that roots should match soils at the same depths. A larger root is almost certainly going to integrate waters from zones larger than the soil surrounding it.

**Reply:** The roots we collected were coarse roots (> 2 mm diameter), as stated in section 3.3 of the Results "Comparison between root water and bulk soil water isotopes at different depths". We have added details regarding root collection and schematic diagram of root excavations (see Figure 2 below), as advised by Reviewer #1. In addition, we have added a brief discussion to the revised manuscript to make the results clearer, as follows:

Regarding root sampling (Page 5 lines 134-144):

*"We excavated a soil cuboid with 160 cm depth, 80 cm width (horizontal distance) and 160 cm length with the main root of the selected tree at the center (Fig. 1d and Fig. 2a). We then divided the cuboid into 64 sub-cuboids (length, 40 cm; width, 40 cm; height, 20 cm) (Fig. 2b) and dug each sub-cuboid one by one to minimize risks of evaporation. 2-3 coarse roots (> 2 mm diameter) from each sub-cuboid were randomly selected and roots from the top few centimeters of the topsoil were not artificially removed. To minimize the influence of attached soil on root water, these sampled roots were rapidly peeled to remove bark, placed in 10 mL vials and sealed with caps then the caps were secured with Parafilm. Finally, these samples were kept in a cool box*

*until storage in the lab at 4℃. To compare the isotopic composition of root and bulk soil water at*

*the same depths, we collected samples of soil around the sampled roots in each sub-cuboid. These*

*soil samples were also rapidly placed in 10 mL vials that were sealed in the same manner as the*

*root samples, then kept in a cool box until storage in the lab at −20℃."*

[Figure]

Figure 2: Schematic diagram of root excavations (a) and measurements (b).

Regarding the brief discussion (Pages 12-13 lines 375-385):

*"We compared the isotopic composition of root water and bulk soil water at the same depth (Fig.*

*6). Contrary to expectations, the root water and bulk soil water at 0-60 cm depths showed*

*consistent $\delta^2H$ and $\delta^{18}O$ values. However, at 80-160 cm depths, $\delta^2H$ and $\delta^{18}O$ values of root water*

*deviated significantly from those of bulk soil water. An alternative explanation for isotopic*

*mismatch at the same depth is that it is due to the complexity of root systems and difficulties in*

*unambiguously determining root traits and functions at specific depths because of the opaque*

*nature of the soil. For example, if collected roots are close to the absorptive roots like fine roots*

*(< 2 mm diameter), they may have similar isotopic composition to bulk soil water at the same*

*depth. In contrast, if they are closer to transport roots like taproots, much of their water content*

*may be from different positions, thereby resulting in inconsistent isotopic composition between*

*root water and surrounding bulk soil water."*

19. 233 I do not understand this "horizontal homogeneity" discussion. Figure 3 shows that

    mobile water and bulk water at a given depth differ. That implies heterogeneity.

**Reply:** As shown in Figure 2 above, we sampled root water for stable isotope analysis at

horizontal distances of 0-40 cm and 40-80 cm from selected tree trunks at the same depth. The

results showed that there were no significant differences ($p > 0.05$) in isotopic composition ($\delta^2H$

and $\delta^{18}O$) of either root water or bulk soil water between 40 cm and 80 cm horizontal distance. To

clarify this, we have revised the sentence in the revised manuscript, as follows (Page 10 lines 291-

293):

*"There were no significant differences ($p > 0.05$) in isotopic composition ($\delta^2H$ and $\delta^{18}O$) of either*

*root water or bulk soil water between 40 cm and 80 cm horizontal distance from selected tree*

*trunks, suggesting that isotopic composition of the soil was horizontally homogenous within 80 cm*

*from tap roots."*

20. 273 So if "separation" here is not the same as the "separation" that has been described

    throughout, it would be useful for the authors to use a more literal descriptor of the process

    that they are investigating.

**Reply:** We have rephrased these sentences, as follows (Page 11 lines 322-329):

*"Gierke et al. (2016) examined the stable isotopic composition of precipitation, bulk soil water*

*and trunk water in a high elevation watershed and their results suggested that mobile water was*

*primarily associated with summer thunderstorms, and thus subject to minimal evaporative loss. In*

*contrast, less mobile water was derived from snowmelt, filling small pores in the shallow soils.*

*Allen et al. (2019) characterized the occurrence of winter and summer precipitation in plant trunk samples using a seasonal origin index and found that winter precipitation was the predominant water source for midsummer transpiration in sampled beech and oak trees. Due to seasonal isotopic cycles in precipitation, there may be clear distinctions in the isotopic composition of mobile water and less mobile water derived from precipitation falling at different times (Bowen et al., 2019)."*

Allen, S.T., Kirchner, J.W., Braun, S., Siegwolf, R.T.W., and Goldsmith, G.R.: Seasonal origins of soil water used by trees, Hydrol. Earth Syst. Sci., 23, 1199-1210, 2019.

Bowen, G.J., Cai, Z.Y., Fiorella, R.P., and Putman, A.L.: Isotopes in the water cycle: regional- to global-scale patterns and applications, Annu. Rev. Earth Planet. Sci., 47, 453-479, 2019.

Gierke, C., Newton, B.T., and Phillips, F.M.: Soil-water dynamics and tree water uptake in the Sacramento Mountains of New Mexico (USA): a stable isotope study, Hydrogeol. J., 24, 805-818, 2016.

21. 291 At this point, I've become overly confused with regard to what the authors consider "TWW" to be.

**Reply:** For reasons stated in our reply to comment 2, we deleted content related to this hypothesis and ecohydrological separation, and paid more attention to the impact of the heterogeneity of soil water on root water uptake in the revised manuscript.

22. 316 What does "mask" mean? Even if there is evaporation, fractionation upon uptake would result in different values between the roots and soils, regardless of the background (soil water) signal.

**Reply:** We have changed the conclusion that the observed isotopic mismatch between root water and bulk soil water at the same depth was caused by isotopic fractionation, as advised by both reviewers (please see our response to main point 3). Thus, we deleted this sentence.

23. 343 The authors should probably specify that this is the duration of the detectable label and not the mean or median of the residence time distribution. Is that correct?

**Reply:** We have amended the sentence as follows (Page 14 lines 442-444):

*"As the time required for isotopic tracer (D$_2$O) to move from the base of a trunk to the upper crown of a tree reportedly ranges from 2.5 to 21 days (Meinzer et al., 2016), the isotopic composition of trunk water may differ from that root water collected on the same day (August 18)."*

24. 345 what is "this interpretation" referring to?

**Reply:** The intended meaning of "This interpretation" was that the root water we collected cannot reflect plant trunk water (xylem water has been changed to trunk water as advised by Reviewer 1) isotopic composition. We have revised the sentence to improve its clarity, as follows (Page 14 lines 442-445):

*"As the time required for isotopic tracer (D$_2$O) to move from the base of a trunk to the upper crown of a tree reportedly ranges from 2.5 to 21 days (Meinzer et al., 2016), the isotopic composition of trunk water may differ from that root water collected on the same day (August 18). We thus measured $\delta^2H$ and $\delta^{18}O$ values of trunk water during our high frequency (ca. 3-day) sampling period from August 4 to September 15, 2019 (Fig. 4)"*

25. 352 It is best to not use "enriched" or "unenriched" unless specifying what they are enriched in (e.g., deuterium) and what they are enriched relative to (e.g., precipitation).

**Reply:** We have clarified these points as follows (Page 15 lines 460-462):

*"Furthermore, previous studies have provided indications that trunk water becomes more enriched in $^{18}O$ due to the temporal declines in sap flow rates (Martin-Gomez et al., 2017) and the mixture of trunk water with leaf water (Brandes et al., 2007)."*

Brandes, E., Wenninger, J., Koeniger, P., Schindler, D., Rennenberg, H., Leibundgut, C., Mayer, H., and Gessler, A.: Assessing environmental and physiological controls over water relations in a Scots pine (*Pinus sylvestris* L.) stand through analyses of stable isotope composition of water and organic matter. Plant Cell Environ., 30,113-127, 2007.

Martin-Gomez, P., Serrano, L., and Ferrio, J.P.: Short-term dynamics of evaporative enrichment of xylem water in woody stems: implications for ecohydrology, Tree Physiol., 37, 511-522, 2017.

26.  356 How is it known that root water is an accurate approach? This is important. It seems like root water tells you a different thing than soil water: the depth of roots contributing to transpiration, rather than the depth of soils contributing to transpiration.

**Reply:** We deleted the calculation of plant water source contributions based on the SIAR model, as advised by Reviewer #1, but kept the conclusion that root water at 100-160 cm depths was the main water source for plants. Although it is difficult to assess the importance of sampled roots for a whole root system's water uptake, root water may reflect the water source of trees better than bulk soil water (which has been more extensively used), for two reasons. First, bulk soil water is commonly collected in cores of 50 cm$^3$ or more (Sprenger et al., 2015; Penna et al., 2018). It is possible to determine the fractions and isotopic composition of bulk soil water held under specific

tension ranges, but information on the spatiotemporal heterogeneity of pore sizes within the cores, and associated effects on uptake patterns, is lost (McCutcheon et al., 2016). Root water is not subject to this deficiency as it consists of water absorbed by fine roots distributed in pores of various sizes. In addition, we systematically collected coarse roots (with > 2 mm diameter) within 80 cm of the main trunk at 20 cm intervals from 0 to 160 cm depths of soil to reduce the potential errors caused by the lack of representativeness of some root water. Our results suggest that trunk water was isotopically closer to root water than bulk soil water. Similarly, measurements of the $\delta^2H$ and $\delta^{18}O$ of bulk soil, trunk and root water from potted *Fagus svlvatica* saplings under control and drought treatments by Barbeta et al. (2020) showed that the $\delta^2H$ of trunk water consistently matched the $\delta^2H$ of root water, and deviated significantly from the $\delta^2H$ of bulk soil water under both treatments.

27. 371 Please be more specific.

**Reply:** We have rephrased this sentence in the revised manuscript, as follows (Page 16 lines 496-498):

*"The presented stable isotope data for bulk soil water, mobile water, less mobile water, root water and trunk water were highly valuable for analyzing the spatial heterogeneity of water fluxes in the root zone, and elucidating the water sources used by the plants."*

---

## Author Response (AR2)

Reply to the comments from the Editor and Reviewers on the manuscript (NO. hess-2020-680) "Insights into isotopic mismatch between soil water and *Salix matsudana* Koidz xylem water from root water isotope measurements".

Dear Editor and Reviewers,

We greatly appreciate the efforts that you and both Reviewers have made to provide further critical feedback regarding our manuscript. We strongly believe that your suggestions have helped us to improve our manuscript. We have carefully considered them and tried to revise the manuscript accordingly, then provided detailed point-by-point replies to your and two reviewers' comments. These comments are presented in blue. Passages changed in specific responses to the comments are presented (together with page and line number) in quotation marks and italic font. All the changes in the revised manuscript have been marked by change-tracking.

**Point-by-point by responses to Editor's comments**

 Please consider rephrasing the end of your introduction, because the paper could still benefit from a more specific primary hypothesis.

**Reply:** Thanks for your suggestion. We have rephrased the hypothesis to make it clearer, as follows (Page 3 lines 73-75):

"We hypothesize that soil water with various mobility is isotopically separated in the soil matrix, which brings about heterogeneity of the soil water, resulting in an isotopic deviation between the measured trunk water and potential water sources of S. matsudana trees during water uptake."

**2. L147: were taken**

**Reply:** We have revised this sentence in the revised manuscript, as follows (Page 4 lines 117-120): "These samples were taken to the laboratory to determine their particle size using a MS 2000 Laser Particle Size Analyzer (Malvern Instruments, Malvern, UK), and to obtain their water retention curves using a CR21G high-speed centrifuge (Hitachi, Japan)."

 L293: "homogeneous" is maybe not the correct expression. I suggest changing it to "bulk soil water had little horizontal variation within 80 cm..."

Reply: We have revised this sentence in the revised manuscript, as suggested (Page 8 lines 226-

227):

"Bulk soil water had little horizontal variation within 80 cm from tap roots."

4. Fig 2: I would suggest to add the schematic plot in the new Fig. 2 to the original Figure 1. There, the schematic graph can replace the map with the Loess Plateau, which is not necessary to show for your study.

**Reply:** Thanks for your suggestion. We have added the schematic plot in the new Figure 2 to the original Figure 1 as follows: